

# The derivative expansion in asymptotically safe quantum gravity: general setup and quartic order

Benjamin Knorr⋆

Perimeter Institute for Theoretical Physics,
31 Caroline Street North, Waterloo, ON N2L 2Y5, Canada

⋆ bknorr@perimeterinstitute.ca

## Abstract

We present a general framework to systematically study the derivative expansion of asymptotically safe quantum gravity. It is based on an exact decoupling and cancellation of different modes in the Landau limit, and implements a correct mode count as well as a regularisation based on geometrical considerations. It is applicable independent of the truncation order. To illustrate the power of the framework, we discuss the quartic order of the derivative expansion and its fixed point structure as well as physical implications.

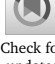

# 1   Introduction

One of the major open problems in fundamental physics is the formulation of a consistent quantum theory of gravity. Despite of several decades of research effort that went into it, no completely consistent and experimentally verified solution exists so far. A big contributor to this status of the field is that quantum gravity effects are expected to be extremely small, and even one-loop effects appear to be unmeasurably tiny at the present resolution of experiments. Thus a lot of guidance must come from theoretical considerations and consistency with Standard Model physics.

A conservative approach to construct a theory of quantum gravity was theorised by Weinberg [1], and goes under the name of Asymptotic Safety. It tries to achieve the quantisation of gravity via postulating quantum scale invariance at high energies induced by a second order phase transition. On the technical level, this translates to an interacting fixed point of the renormalisation group flow. Results obtained in $2 + \epsilon$ dimensions indeed suggest the existence of such a fixed point, at least near two dimensions [2–7], but the extrapolation to the physical case of four dimensions remained difficult for some time.

With the advent of modern, non-perturbative functional renormalisation group techniques [8–10], Asymptotic Safety picked up speed again. The seminal paper by Reuter [11] indeed showed evidence for the existence of a fixed point in four dimensions within a minimal approximation. Since then, a growing body of work [12–54] based on evermore improved approximations solidifies the picture, even when matter is included [55–73]. Phenomenological applications have been discussed in [74–91]. For reviews of the field, see [92–98], and for a critical discussion of open problems, see [99, 100]. Lattice formulations like Euclidean or Causal Dynamical Triangulations [101–109] indicate the existence of a second order phase transition as well.

One systematic way to study the stability of these results is the derivative expansion. In this, interactions which include up to a set amount of derivatives acting on the fundamental field are taken into account. In the context of gravity, this corresponds to powers of curvature tensors and their covariant derivatives. Surprisingly, to this point a complete non-perturbative discussion of the fourth order approximation has not been carried out in the context of Asymptotic Safety. This has both conceptual and technical reasons. Conceptually, the functional renormalisation group relies on the choice of a regulator. The regularisation of operators in a curved spacetime is much more involved than in a flat spacetime. Only partial results have been obtained regarding this problem, and most rely on the particular structure of the theory at second order in derivatives. On the technical side, the computational complexity increases tremendously with the approximation order.

In this paper, we provide a solution to the problem of regularisation in asymptotically safe quantum gravity motivated by geometrical arguments. It is based on the decomposition into

gauge variant and invariant components of the field, and can be applied to any order of the derivative expansion, including resummations in terms of form factors. We also provide new technical insights that allow for a generic implementation of the computation of the renormalisation group running. To illustrate the framework, we compute the complete non-perturbative renormalisation group running in quantum gravity to fourth order in the derivative expansion.

The main results that we obtain in the concrete computation are as follows:

- Fourth order gravity admits a non-perturbative fixed point.

- Spacetimes with a negative (positive) Euler characteristic dominate (are suppressed in) the Euclidean path integral of Asymptotic Safety, whereas spacetimes with vanishing Euler characteristic contribute with unit strength.

- While the inclusion of the square of the Weyl tensor provides a well-controlled extension of the Reuter fixed point, the squared Ricci scalar introduces some kind of instability into the results. This has been observed before, and the inclusion of higher order terms seems to stabilise the system [19, 21, 23, 29, 32, 34, 36, 39, 43, 47, 51–53, 110].

This paper is structured as follows. In section 2 we briefly discuss the functional renormalisation group (FRG) which lies at the heart of our investigations of Asymptotic Safety. In particular, we define a set of criteria that well-behaved regulators and flows should satisfy. Section 3 discussed the decomposition of fields into gauge variant and invariant components. We first provide a simpler discussion in the context of an Abelian gauge field, and then investigate how much we can transfer the structure to the gravitational case. This discussion leads to our proposal of a well-motivated regularisation scheme in gravity. Sections 4, 5 and 6 as well as appendix A collect some technical machinery that allow us to decrease the technical complexity to a manageable level. The setup is illustrated in section 7, where we carry out the computation to fourth order in derivatives. We then conclude and provide an outlook in section 8.

## 2 Functional renormalisation group

The tool that we will use to investigate the non-perturbative renormalisation group flow is the functional renormalisation group. Its central object is the effective average action $\Gamma_k$, which interpolates between a microscopic action $S$ in the limit $k \to \infty$, and the standard quantum effective action $\Gamma$ at $k = 0$. Decreasing the fiducial scale $k$ then corresponds to integrating out modes in the Wilsonian sense. The dependence of $\Gamma_k$ on $k$ is governed by the following functional integro-differential equation [8–10]:

$$\partial_t \Gamma_k = \frac{1}{2} \text{STr}\left[ \left( \Gamma_k^{(2)} + \mathfrak{R}_k \right)^{-1} \partial_t \mathfrak{R}_k \right]. \tag{1}$$

In this, $\partial_t = k \partial_k$ is the logarithmic scale derivative, $\Gamma_k^{(2)}$ denotes the second functional derivative of $\Gamma_k$, $\mathfrak{R}_k$ is a regulator term which acts as a momentum-dependent mass, and the supertrace STr indicates a sum over discrete (*e.g.* spacetime or gauge bundle), and an integral over continuous (*e.g.* momentum) indices. For reviews of the FRG see [97, 111–115].

From the flow equation (1) we can extract the beta functions of a theory, *i.e.*, (integro-)differential equations that govern the scale dependence of couplings. Typically, one discusses them for the dimensionless versions of couplings, where we multiply the coupling with a power of the scale $k$ to make it dimensionless. Combined zeros of all beta functions are called fixed points. We will indicate fixed point values of couplings by an asterisk. If all couplings vanish at a given fixed point, it is called Gaussian, otherwise we call it interacting or non-Gaussian.

A fixed point is then characterised by its critical exponents, which describe the linearised flow around it. They are defined as minus the eigenvalues of the matrix of partial derivatives of the beta functions with respect to the couplings, evaluated at the fixed point. Positive (negative) critical exponents indicate relevant (irrelevant) directions, so that the flow is attracted towards (repelled away from) the fixed point when increasing the scale $k$. To end up with a predictive theory, a fixed point can only have finitely many positive critical exponents, since they correspond to the number of independent measurements that one has to perform to fix the theory uniquely. An asymptotically safe fixed point is then defined as an interacting fixed point with a finite number of positive critical exponents.

In quantum gravity, we have to add a gauge fixing term to the action to make the propagator that appears in the flow equation well-defined. On the practical level, this is implemented with the help of the background field method. We split the metric $g$ into an arbitrary background metric $\bar{g}$ and (not necessarily small) fluctuations $h$ around it,

$$g_{\mu\nu} = \bar{g}_{\mu\nu} + h_{\mu\nu}. \tag{2}$$

Other (non-linear) ways to perform this separation have been investigated, see *e.g.* [28, 30, 32, 33, 39, 42, 51, 57, 116–122] for examples in the context of Asymptotic Safety. Such a split is also necessary to define the regulator term. The advantage of this method is that invariance with respect to background diffeomorphisms can be maintained in every step. However, as a downside, the regulator and the gauge fixing term break certain Ward identities, so that in principle one has to deal with modified Ward identities that have to be fulfilled together with the flow equation. In this work, we will focus on a background field approximation, which neglects these issues and corresponds to setting the fluctuation field $h$ to zero after taking the second variation. For an in-depth discussion of these issues, see *e.g.* [15, 18, 23–25, 27, 38, 40, 41, 46, 49, 50, 56, 59, 67, 71, 98, 122–140].

In practice, we generally have to make approximations to solve the flow equation (1). Only in special cases, an exact solution is possible, see [141]. For recent progress in constructing exact solutions to the flow equation, see also [142]. In the following, we will discuss a (covariant) derivative expansion of the effective average action. The order of the expansion is then the maximal number of derivatives that act on the metric.

A generic problem in this setup is the systematic choice of a regulator. Partial results have been obtained in the literature, notably [12], but they typically rely on technical assumptions that potentially do not carry over to higher orders, or only at considerable technical cost. Two of the goals of this paper are to establish general criteria that a good flow should have, and a generic way to choose a regulator which gives rise to a good flow. Before we enter this discussion, we briefly note that in this paper we consider a Euclidean flow. Results with Lorentzian signature can be found in [13, 37, 100, 117, 143–148].

## 2.1 Criteria for a successful flow

Having introduced the machinery, we will now discuss some criteria that we expect a flow to have. The first condition is what we call a correct mode count. The idea is that the flow of the cosmological constant should have a very generic form, and effectively counts the number of physical degrees of freedom. For example, for a free scalar field, the contribution to the flow of the cosmological constant is

$$\frac{1}{2} \frac{1}{(4\pi)^{\frac{d}{2}}} \frac{1}{\Gamma\left(\frac{d}{2}\right)} \int_0^\infty \mathrm{d}y \, y^{\frac{d}{2}-1} \frac{\partial_t \mathfrak{R}_k(y)}{y + \mathfrak{R}_k(y)}. \tag{3}$$

In this, the factor of a half comes directly from the flow equation (1), and the rest of the prefactor of the integral, as well as its measure $y^{\frac{d}{2}-1}$, come from the heat kernel. The integrand

is then just the product of the regularised propagator and the scale derivative of the regulator. We thus demand that the contribution of any physical degree of freedom comes in this form: an integral over the *full* regularised propagator with the prefactors as above.

A second criterion that we want to implement is that the flow should admit a finite Landau gauge limit. It corresponds to a strict implementation of the gauge condition, and is a fixed point of the flow [41, 149]. This puts indirect constraints on the regulator as well, which are however hard to spell out concretely. A well-motivated setup with an a priori guaranteed finite Landau limit will be presented below. In any case, this can be easily checked a posteriori, once the flow is computed.

A third criterion concerns the choice of operator that is regularised, and the tensor structure of the regulator. As a convention, we will always assume that the operator is normalised such that the (principal) Laplace part of the operator comes with a unit prefactor. We then require that the regularised operator, and the tensor structure of the regulator, have a physical or mathematical motivation. We will discuss this more concretely in the next section.

# 3 Field decompositions in curved spacetime

To solve the problem of finding a suitable regularisation, we will now discuss the decomposition of fields into components. A guiding principle will be that we rescale the modes such that no non-trivial Jacobians are introduced into the path integral. As a simple example, we will start with an Abelian gauge field, for which we can easily derive all necessary ingredients. We then make a short digression to discuss more general vector fields, and discuss their regularisation. This includes the gravitational Faddeev-Popov ghost, which has a slightly different structure than an Abelian gauge field. Finally, we will discuss the mode decomposition of the graviton and the regularisation strategy that this suggests.

Let us mention that in practice, it is easier to avoid working with decomposed fields when it comes to computing functional traces like in (1). We will provide an argument for this in subsection 3.4. The aim of this section is to motivate our choice of regularisation from a geometric perspective. In concrete computations, we then implement the regularisation on the level of the full fields with suitable projectors so that we can use standard heat kernel techniques. We will nevertheless provide all details in the hope that some readers with different applications might find them useful.

## 3.1 Transverse decomposition of an Abelian gauge field

We will now discuss the decomposition of an Abelian gauge field into modes, and their respective regularisation. The starting point will be the gauge fixing condition, from which we derive the decomposition. We then rescale some of the fields to eliminate the need of Jacobians. After a discussion of the projectors onto the different modes, we discuss the natural operator that arises from the decomposition, including its heat kernel properties. This will suggest a particular way to regularise the theory. Finally, we will make some comments on the origin of the natural operator, and briefly discuss how to regularise higher derivative Abelian theories.

**Gauge fixing condition**    The standard choice for a linear, covariant gauge fixing is

$$\mathfrak{F}^{\mu}A_{\mu} = D^{\mu}A_{\mu}\,, \tag{4}$$

where $D$ is the covariant derivative. The sought-after decomposition should be such that only the gauge mode appears in this expression, *i.e.*, the physical mode of the decomposition is annihilated by $\mathfrak{F}$. In this case, it means that the physical mode is (covariantly) transverse.

**Transverse decomposition**   By this argument, the decomposition into gauge invariant and gauge variant modes is the well-known decomposition into transverse and longitudinal components,

$$A_\mu = A_\mu^{\mathrm{T}} + \mathfrak{F}_\mu \tilde\phi \equiv A_\mu^{\mathrm{T}} + D_\mu \tilde\phi \,, \qquad D^\mu A_\mu^{\mathrm{T}} = 0 \,. \tag{5}$$

Here, $A_\mu^{\mathrm{T}}$ is the gauge invariant transverse mode and $\tilde\phi$ a scalar field underlying the longitudinal gauge variant component. In consequence, acting with the gauge fixing operator on the field gives

$$\mathfrak{F}^\mu A_\mu = D^\mu D_\mu \tilde\phi = -\Delta \tilde\phi \,, \tag{6}$$

where we introduced the Laplacian

$$\Delta = -D^\mu D_\mu \,. \tag{7}$$

At this point, let us mention that there is still a potential gauge redundancy in the transverse mode. It can be shifted by the derivative of a solution to the Laplace equation,

$$A_\mu^{\mathrm{T}} \mapsto A_\mu^{\mathrm{T}} + D_\mu \omega \,, \qquad \Delta \omega = 0 \,. \tag{8}$$

We will not address this problem in this work.

**Jacobian**   When calculating an integral, every variable transformation gives rise to a Jacobian. This is also the case when we want to calculate a path integral and perform the decomposition (5). The arising Jacobian can be calculated by a standard trick [150]. We can consider the exponential integral

$$\int \mathcal{D}A \, e^{-\int d^d x \sqrt{g} A_\mu A^\mu} \simeq \int \mathcal{D}A^{\mathrm{T}} \mathcal{D}\tilde\phi \, \mathcal{J}_A \, e^{-\int d^d x \sqrt{g}\left[A_\mu^{\mathrm{T}} A^{\mathrm{T}\mu} + \phi \Delta \phi\right]} \simeq \mathcal{J}_A (\det \Delta)^{-1/2} \,. \tag{9}$$

Since the overall normalisation of the path integral is inessential, we neglected overall factors. We also dropped boundary terms upon integration by parts. The Jacobian has to be chosen to cancel the determinant, so that

$$\mathcal{J}_A = (\det \Delta)^{1/2} \,. \tag{10}$$

**Avoiding the Jacobian**   We would like to avoid the introduction of such determinants, and formulate a decomposition which has field components of the same mass dimension. This is not the case for (5) - the scalar $\tilde\phi$ has a relative mass dimension of one unit less in comparison to $A$ and $A^{\mathrm{T}}$. In this case, the solution is straightforward: we define a new scalar field

$$\phi = \frac{1}{\sqrt{\Delta}} \tilde\phi \,. \tag{11}$$

This is well-defined as long as we exclude potential negative or zero modes of the Laplacian. With this definition, the new decomposition of the vector field

$$A_\mu = A_\mu^{\mathrm{T}} + D_\mu \frac{1}{\sqrt{\Delta}} \phi \,, \tag{12}$$

does not give rise to a Jacobian. A path integral over $A$ is thus the same as a path integral over $A^{\mathrm{T}}$ and $\phi$, up to the aforementioned subtleties of individual modes.

**Projectors**  Let us now write down the projectors onto the transverse and longitudinal components. It is easy to see that

$$\Pi^{\mathrm{L}}{}_\mu{}^\nu = -D_\mu \frac{1}{\Delta} D^\nu, \qquad \Pi^{\mathrm{T}}{}_\mu{}^\nu = \delta_\mu{}^\nu - \Pi^{\mathrm{L}}{}_\mu{}^\nu, \tag{13}$$

project onto the longitudinal and transverse components,

$$\left(\Pi^{\mathrm{L}} A\right)_\mu = D_\mu \frac{1}{\sqrt{\Delta}} \phi, \qquad \left(\Pi^{\mathrm{T}} A\right)_\mu = A_\mu^{\mathrm{T}}. \tag{14}$$

Anticipating the discussion for the graviton case, note that the projector onto the gauge mode is entirely built up from the gauge fixing operator, as we can write

$$\Pi^{\mathrm{L}}{}_\mu{}^\nu = \mathfrak{F}_\mu \frac{1}{\mathfrak{F}^\alpha \mathfrak{F}_\alpha} \mathfrak{F}^\nu. \tag{15}$$

This might seem trivial in the Abelian case, but the structure is true more generally, and a consequence of demanding that different modes are orthogonal to each other.

**Natural operator**  In connection with the projection operators, we will introduce the concept of the "natural" operator associated with the field, $\Delta_A$. We define it as an operator of Laplace type which commutes with the projectors, and has a compatible index structure such that it maps a given field to a field with the same index structure. The principal part of the operator is then normalised to one. For a vector field the operator $\Delta_A$ can be constructed easily. Observe that

$$\left(\Delta \delta_\mu{}^\nu + R_\mu{}^\nu\right) \Pi^{\mathrm{L}}{}_\nu{}^\rho A_\rho = -D_\mu D^\rho A_\rho = \Pi^{\mathrm{L}}{}_\mu{}^\nu \left(\Delta \delta_\nu{}^\rho + R_\nu{}^\rho\right) A_\rho, \tag{16}$$

so that

$$\Delta_{A\mu}{}^\nu = \Delta \delta_\mu{}^\nu + R_\mu{}^\nu, \qquad \left[\Delta_A, \Pi^{\mathrm{T}}\right] = \left[\Delta_A, \Pi^{\mathrm{L}}\right] = 0, \tag{17}$$

is the sought-after operator. When calculating the renormalisation group running of couplings, using this operator simplifies calculations. The above operator is the unique local, Laplace-type operator whose set of eigenfunctions splits into transverse and longitudinal eigenfunctions.

**Heat kernel coefficients of $\Delta_A$**  Let us illustrate the special role of the operator $\Delta_A$ by considering its heat kernel coefficients. While it has been found that the heat kernel coefficients of a pure Laplace operator in the space of transverse functions is singular in even dimensions [151], we will illustrate now that this is not the case for the natural operator (17). To that extent, we consider the (traced) heat kernel coefficients in the longitudinal sector,

$$H_1^{\mathrm{L}}(\Delta_A) = \mathrm{Tr}\, \Pi^{\mathrm{L}} e^{-s\Delta_A} = -\mathrm{Tr} D_\mu D^\nu \left(\frac{e^{-s\Delta_A}}{\Delta_A}\right)_\nu{}^\rho = -\int_0^\infty \mathrm{d}t\, \mathrm{Tr}\, D_\mu D^\nu \left(e^{-(s+t)\Delta_A}\right)_\nu{}^\rho. \tag{18}$$

The trace can be calculated with standard off-diagonal heat kernel techniques [151, 152]. One finds that the heat kernel coefficients agree precisely with the heat kernel coefficients corresponding to a pure Laplacian acting on a scalar,

$$H_1^{\mathrm{L}}(\Delta_A) = H_0(\Delta). \tag{19}$$

This should not come as a surprise - the gauge fixing operator equips the longitudinal scalar $\phi$ with a plain Laplacian,

$$\int \mathrm{d}^d x\, \sqrt{g} \left(\mathfrak{F}^\mu A_\mu\right)^2 = \int \mathrm{d}^d x\, \sqrt{g}\, \phi \Delta \phi. \tag{20}$$

One different way to interpret (19) is that the vector version of the Laplacian acting on a scalar is the operator $\Delta_A$.

From this, we can calculate the transverse heat kernel,

$$H_1^{\mathrm{T}}(\Delta_A) = \mathrm{Tr}\,\Pi^{\mathrm{T}}\,e^{-s\Delta_A} = \mathrm{Tr}\,e^{-s\Delta_A} - H_1^{\mathrm{L}}(\Delta_A). \tag{21}$$

The total contribution of a free Abelian gauge field in curved spacetime is thus

$$H_1(\Delta_A) = H_1^{\mathrm{T}}(\Delta_A) + H_1^{\mathrm{L}}(\Delta_A) = \mathrm{Tr}\,e^{-s\Delta_A}. \tag{22}$$

This seems like a trivial statement - the contribution of a vector is the sum of the contributions of the individual modes. In a quantum field theory setting, where regularisation is necessary, this becomes a guiding principle. Only those regularisations that preserve this additive structure are proper. In particular, if one were to regularise only the flat part of the operator, this sum rule is violated, and heat kernel coefficients diverge in even dimensions [151]. This finding has particular relevance for the flow equation. For a classification of different regulator types see [153].

We thus can finally formulate our regularisation strategy for an Abelian vector field. The transverse part is regularised using the operator $\Delta_A$, so that[1]

$$\mathfrak{R}_k^{\mathrm{T}}(\Delta_A) = k^2 \mathcal{R}_k^{\mathrm{T}}(\Delta_A/k^2)\Pi^{\mathrm{T}}. \tag{23}$$

In the longitudinal sector, we can either also adapt this operator if in practice we work without the explicit decomposition,

$$\mathfrak{R}_k^{\mathrm{L}}(\Delta_A) = k^2 \mathcal{R}_k^{\mathrm{L}}(\Delta_A/k^2)\Pi^{\mathrm{L}}, \tag{24}$$

or equivalently, we use a pure Laplacian if we use the decomposition,

$$\mathfrak{R}_k^{\phi}(\Delta) = k^2 \mathcal{R}_k^{\mathrm{L}}(\Delta/k^2). \tag{25}$$

In practice, it is useful to expand the regulator in powers of curvature. This can be done easily, see *e.g.* [154]. This is possibile even within a form factor setup, see *e.g.* appendix C of [139].

**General actions and identifying propagators and interactions**   Coming back to a more general scope, the precise shape of the operator $\Delta_A$ is no coincidence. The action of a free Abelian gauge field is proportional to the square of the field strength

$$F_{\mu\nu} = D_\mu A_\nu - D_\nu A_\mu. \tag{26}$$

The corresponding two-point function of a free Abelian gauge field reads

$$\Delta\delta_\mu{}^\nu + D_\mu D^\nu = \left(\Delta_A\Pi^{\mathrm{T}}\right)_\mu{}^\nu, \tag{27}$$

which is precisely the natural operator found above, together with a projector onto the transverse mode. This observation has a profound consequence for the definition of the nonperturbative Abelian gauge field propagator in curved spacetime. The most general term quadratic in the Abelian gauge field strength can be written as

$$\int \mathrm{d}^d x\,\sqrt{g}\,F^{\mu\nu}E_{\mu\nu}{}^{\rho\sigma}F_{\rho\sigma}, \tag{28}$$

---

[1]We generally take the convention that $\mathfrak{R}_k$ has a dimension of $k^2$, whereas $\mathcal{R}_k$ is dimensionless. In addition, the latter is also defined to have a dimensionless argument. This gives rise to various factors of $k^2$ in some of the equations.

where $E$ is some operator which is independent of $A$. If we want to preserve the above structure, namely that this action gives rise to a two-point function of the transverse Abelian gauge field of the form[2]

$$\left(e(\Delta_A)\Delta_A\Pi^{\mathrm{T}}\right)_\mu{}^\nu,\tag{29}$$

with an arbitrary function $e$ which defines Abelian gauge field propagation in curved spacetime, we have to chose

$$E_{\mu\nu}{}^{\rho\sigma} = e\left(\Delta\delta_{[\mu}{}^{[\rho}\delta_{\nu]}{}^{\sigma]} + 4R_{[\mu}{}^{[\rho}\delta_{\nu]}{}^{\sigma]} - 2R_{\mu\nu}{}^{\rho\sigma}\right).\tag{30}$$

The operator in brackets can be derived by demanding

$$-D^\mu E_{\mu\nu}{}^{\rho\sigma}F_{\rho\sigma} \propto (e(\Delta_A)\Delta_A A)_\nu,\tag{31}$$

together with the assumption that it is a local Laplace-type operator. The above gives a unique prescription of how to split the action of an Abelian gauge field in an arbitrarily curved spacetime into propagator and interaction terms: first collect the pieces that survive in the flat spacetime limit, then complete the operator to $E$ to lift it into curved space. This represents a minimally coupled Abelian gauge field with a non-trivially momentum-dependent propagator. Any term with two Abelian gauge field strengths and some power of curvature that is not of this form is then a genuine interaction term. Note that the regularisation prescription that we outlined above is still applicable.

## 3.2 General vector field

Before we continue with the case of the graviton, let us briefly discuss the regularisation of more general vector fields. This will also cover the gravitational Faddeev-Popov ghost. A general second order two-point function for a vector looks like

$$\boldsymbol{\Delta}_\mu{}^\nu = \Delta\delta_\mu{}^\nu + bD_\mu D^\nu + \tilde{\mathcal{E}}_\mu{}^\nu.\tag{32}$$

Here $b$ is a number and $\tilde{\mathcal{E}}$ is a multiplicative operator (often referred to as endomorphism). For the gravitational Faddeev-Popov ghost, and with the gauge fixing (41) defined below, we have

$$b = 2\frac{1+\beta}{d} - 1, \qquad \tilde{\mathcal{E}}_\mu{}^\nu = -R_\mu{}^\nu.\tag{33}$$

For later reference, we will call this ghost operator $\Delta_c$,

$$\Delta_{c\mu}{}^\nu = \Delta\delta_\mu{}^\nu + \left(2\frac{1+\beta}{d} - 1\right)D_\mu D^\nu - R_\mu{}^\nu.\tag{34}$$

Let us rewrite the operator $\boldsymbol{\Delta}$ in terms of the operator $\Delta_A$:[3]

$$\boldsymbol{\Delta} = \Delta_A\left(\Pi^{\mathrm{T}} + (1-b)\Pi^{\mathrm{L}}\right) + \mathcal{E},\tag{35}$$

---

[2]Formally, we define a function of an operator by either its Taylor series, or an inverse integral transform of exponential type, *e.g.*, an inverse Laplace transform. This covers most interesting functions. In particular, it includes the logarithm via

$$\ln x = \int_0^\infty \mathrm{d}s\, \frac{e^{-s} - e^{-sx}}{s}.$$

To prove some formulas, we will work with inverse Laplace transforms. All manipulations that we perform here and later in the paper will however also go through without significant changes for functions like the logarithm.

[3]In this form we see that for the special case $b = 1$, the kinetic part of the operator is proportional to a projector, and thus not invertible in a derivative expansion. For gravity, this corresponds to the singular gauge fixing $\beta = d-1$, as has been noted before [28]. We will see the imprint of this in some parts of the trace (38) below.

where we introduced the shifted endomorphism

$$\mathcal{E}_\mu{}^\nu = \tilde{\mathcal{E}}_\mu{}^\nu - R_\mu{}^\nu. \tag{36}$$

Due to the central role that is played by $\Delta_A$, we will still use it as the operator in our regulator choice. This means that we treat $\mathcal{E}$ as the curvature interaction term. Concretely, the regularised version of (35) reads

$$\mathbf{\Delta}^{\mathrm{reg}} = \left(\Delta_A + k^2 \mathcal{R}_k(\Delta_A/k^2)\right)\left(\Pi^{\mathrm{T}} + (1-b)\Pi^{\mathrm{L}}\right) + \mathcal{E}. \tag{37}$$

For simplicity, we chose the same regulator shape function in both sectors. This regularisation satisfies the mode count requirement: the contribution of a vector trace in the flat limit counts as $d$ modes, independent of the value of $b$. As a matter of fact, for $\mathcal{E} = 0$ the dependence on $b$ drops out. This is reasonable since when we decompose the vector into transverse and longitudinal parts, we still have to canonically normalise the field $\phi$. The rescaling then eliminates all occurrences of $b$. The above regulator choice implements this idea in the presence of a finite endomorphism. In turn, the flow equation automatically takes care of the above-mentioned rescaling if we regularise like in (37).

With standard off-diagonal heat kernel methods, one can then derive the contribution of the trace of a vector regularised in such a way within the FRG in a derivative expansion. In general dimension $d$, it reads

$$
\begin{aligned}
\mathfrak{T}_1 &= \frac{1}{2}\mathrm{STr}\left[(\mathbf{\Delta}^{\mathrm{reg}})^{-1}\left(\Pi^{\mathrm{T}} + (1-b)\Pi^{\mathrm{L}}\right)\partial_t[k^2\mathcal{R}_k(\Delta_A/k^2)]\right] \\
&= \frac{1}{2}\mathrm{STr}\left[\sum_{n\geq 0}\left(-\frac{1}{\Delta_A + k^2\mathcal{R}_k(\Delta_A/k^2)}\left(\Pi^{\mathrm{T}} + \frac{1}{1-b}\Pi^{\mathrm{L}}\right)\mathcal{E}\right)^n \frac{\partial_t[k^2\mathcal{R}_k(\Delta_A/k^2)]}{\Delta_A + k^2\mathcal{R}_k(\Delta_A/k^2)}\right] \\
&\simeq \frac{1}{2}\frac{1}{(4\pi)^{\frac{d}{2}}}\frac{1}{\Gamma\left(\frac{d}{2}\right)}\int\sqrt{g}\Bigg[d\,\mathcal{I}_1^1 + \frac{d-2}{2}\left(\frac{d}{6}-1\right)R\mathcal{I}_1^2 - \left(1 + \frac{1}{d}\frac{b}{1-b}\right)\mathcal{E}_\mu{}^\mu\mathcal{I}_2^1 \\
&\quad + (d-2)(d-4)\left(\frac{d-12}{288}R^2 - \frac{d-90}{720}R_{\mu\nu}R^{\mu\nu} + \frac{d-15}{720}R_{\mu\nu\rho\sigma}R^{\mu\nu\rho\sigma}\right)\mathcal{I}_1^3 \\
&\quad + \left(\left(\frac{d-2}{2} + \frac{1}{6}\frac{b}{1-b}\right)R^{\mu\nu}\mathcal{E}_{\mu\nu} - \frac{1}{12}\left((d-2) + \frac{b}{1-b}\right)R\mathcal{E}_\mu{}^\mu\right)\mathcal{I}_2^2 \\
&\quad + \left(\left(1 + \frac{2}{d}\frac{b}{(1-b)^2}\left(1 - \frac{d+1}{d+2}b\right)\right)\mathcal{E}^{\mu\nu}\mathcal{E}_{\mu\nu} + \frac{1}{d(d+2)}\frac{b^2}{(1-b)^2}\mathcal{E}_\mu{}^\mu\mathcal{E}_\nu{}^\nu\right)\mathcal{I}_3^1\Bigg].
\end{aligned}
\tag{38}
$$

In the last line we neglected terms with more than four derivatives, and we introduced the integrals

$$\mathcal{I}_m^n = 2\,k^{d-2(n+m-2)}\int_0^\infty \mathrm{d}z\, z^{\frac{d}{2}-n}\frac{\mathcal{R}_k(z) - z\mathcal{R}_k'(z)}{(z + \mathcal{R}_k(z))^m}. \tag{39}$$

Higher orders can be calculated systematically. As expected from the general form of the trace, terms with $m$ powers of $\mathcal{E}$ come with $(m+1)$ powers of the propagator in the integrals. From the prefactor of the integral of the volume term, we can explicitly see that the mode count of $d$ modes for a vector is implemented correctly.

Since later we are interested in $d = 4$, we have to be careful in the evaluation of $(d-4)\mathcal{I}_1^3$. One can show that

$$\lim_{d\to 4}(d-4)\mathcal{I}_1^3 = 4. \tag{40}$$

## 3.3 Decomposition of the graviton

We will now turn our attention to the decomposition of the graviton. In doing so, we will try to follow the same steps as for the Abelian gauge field. As it turns out, much of the construction

can be done in a similar way, but there are some key differences. From the general theory of irreducible representations, we anticipate a rank two transverse traceless tensor, a transverse vector and two mixing scalars. The scalars can be diagonalised with respect to the gauge fixing, so that one linear combination is gauge invariant, while the other is gauge variant, and will be recombined with the pure gauge transverse vector. In the construction of the decomposition, we took inspiration from [150], but with view on our goal of defining a useful regularisation and avoiding Jacobians, our implementation differs in some details.

In the continuum approach to quantum gravity, the use of the background field method is hard to avoid. Thus, as indicated earlier, in the discussion below we will make use of it and construct the decomposition with respect to background quantities, indicated by an overbar. For an alternative approach towards defining a decomposition with respect to the full metric, see [23].

**Gauge fixing condition**    Again we start by specifying a gauge condition. For gravity one typically considers the one-parameter family of linear covariant gauges

$$\mathfrak{F}_\alpha{}^{\mu\nu} = \delta_\alpha{}^{(\mu}\bar{D}^{\nu)} - \frac{1+\beta}{d}\bar{g}^{\mu\nu}\bar{D}_\alpha\,, \tag{41}$$

where $\beta$ is a gauge parameter that determines the way of how the two scalar modes mix to give the gauge invariant and the gauge variant scalar mode.

**Transverse traceless decomposition**    We make the following ansatz for the decomposition of the metric fluctuation:

$$h_{\mu\nu} = h_{\mu\nu}^{\mathrm{TT}} + 2\mathfrak{F}^\alpha{}_{\mu\nu}\zeta_\alpha + \mathfrak{Q}_{\mu\nu}\theta\,, $$
$$\bar{D}^\mu h_{\mu\nu}^{\mathrm{TT}} = 0\,, \qquad \bar{g}^{\mu\nu}h_{\mu\nu}^{\mathrm{TT}} = 0\,, \qquad \mathfrak{Q}_{\mu\nu} = \mathfrak{Q}_{(\mu\nu)}\,. \tag{42}$$

Here $h^{\mathrm{TT}}$ is the transverse traceless mode, which is the gauge invariant spin two mode. The pure gauge vector $\zeta$ is introduced by means of the gauge fixing operator, similar to the Abelian case. We do not decompose it further into transverse and longitudinal components. The scalar $\theta$ is then the gauge invariant scalar.

By construction, $h^{\mathrm{TT}}$ is annihilated by the gauge operator,

$$\mathfrak{F}_\alpha{}^{\mu\nu}h_{\mu\nu}^{\mathrm{TT}} = 0\,. \tag{43}$$

We also require that the gauge condition also annihilates $\theta$,

$$\mathfrak{F}_\alpha{}^{\mu\nu}\mathfrak{Q}_{\mu\nu}\theta = 0\,. \tag{44}$$

Let us construct the operator $\mathfrak{Q}$. Observe that for the choice $\beta = 0$, the gauge fixing operator is traceless, so that the gauge invariant scalar mode is the trace. This motivates the ansatz

$$\mathfrak{Q}_{\mu\nu} = \bar{g}_{\mu\nu} - 2\beta Q_{\mu\nu}\,, \tag{45}$$

where $Q$ is symmetric. In the following we will assume that there is no term proportional to the metric in $Q_{\mu\nu}$. If there were such a term, we could pull it out and rescale the field $\theta$ to enforce a unit coefficient as in the above equation.

Acting with the gauge fixing operator on this gives

$$\mathfrak{F}_\alpha{}^{\mu\nu}\mathfrak{Q}_{\mu\nu}\theta = -\beta\left[\bar{D}_\alpha + 2\bar{D}^\mu Q_{\mu\alpha} - 2\frac{1+\beta}{d}\bar{D}_\alpha Q^\mu{}_\mu\right]\theta = 0\,. \tag{46}$$

Let us now assume that $\beta \neq 0$ so that we can fix the operator $Q$. We can rewrite the equation as

$$2\bar{D}_\mu \left[ Q^\mu{}_\alpha - \frac{1+\beta}{d} \delta_\alpha{}^\mu Q^\nu{}_\nu \right] \theta = -\bar{D}_\alpha \theta \,. \tag{47}$$

This equation tells us that acting with a derivative from the left on the operator $Q$ and contracting should essentially give back the derivative, *i.e.* it is some kind of longitudinal projector, but with an unusual index structure. For that reason, let us make the ansatz

$$Q_{\mu\nu} = \bar{D}_{(\mu} \mathcal{X}_{\nu)}{}^\alpha \bar{D}_\alpha \,, \tag{48}$$

and derive the form of $\mathcal{X}$ so that the above equation is fulfilled. Note again that we could add a term proportional to the background metric to $Q$, but this would only yield a total rescaling of $\mathfrak{Q}$, as discussed above. With (48), we get

$$
\begin{aligned}
-\bar{D}_\mu \theta &= \left[ \bar{D}^\alpha \bar{D}_\alpha \mathcal{X}_\mu{}^\nu + \bar{D}^\alpha \bar{D}_\mu \mathcal{X}_\alpha{}^\nu - 2\frac{1+\beta}{d} \bar{D}_\mu \bar{D}^\alpha \mathcal{X}_\alpha{}^\nu \right] \bar{D}_\nu \theta \\
&= -\left[ \bar{\Delta} \delta_\mu{}^\alpha - \bar{D}^\alpha \bar{D}_\mu + 2\frac{1+\beta}{d} \bar{D}_\mu \bar{D}^\alpha \right] \mathcal{X}_\alpha{}^\nu \bar{D}_\nu \theta \,.
\end{aligned}
\tag{49}
$$

We conclude that $\mathcal{X}$ must be the inverse of the operator in the brackets,

$$\mathcal{X} = \left[ \bar{\Delta} \delta_\cdot{}^\cdot - \bar{D}^\cdot \bar{D}_\cdot + 2\frac{1+\beta}{d} \bar{D}_\cdot \bar{D}^\cdot \right]^{-1} \,. \tag{50}$$

In fact, this is precisely the kinetic operator of the Fadeev-Popov ghosts (34) associated to the gauge fixing operator (41),

$$\mathcal{X} = \Delta_c^{-1} \,. \tag{51}$$

This means that, up to the condition that

$$\beta < d - 1 \,, \tag{52}$$

which is needed for positivity and invertibility, see (35), the operator and its inverse should exist inside the first Gribov region.

**Jacobian** Having derived the decomposition into gauge invariant and gauge variant modes, let us calculate the Jacobian that arises from this variable transformation. We consider the same integral as for the case of the Abelian gauge field. Before we do that, we first define the operator

$$\mathfrak{Q}^{\dagger\mu\nu} = \bar{g}^{\mu\nu} - 2\beta \bar{D}^\alpha \mathcal{X}_\alpha{}^{(\mu} \bar{D}^{\nu)} \,, \tag{53}$$

which fulfils

$$\int \mathrm{d}^d x \sqrt{\bar{g}} \, Y \, \mathfrak{Q}_{\mu\nu} \theta = \int \mathrm{d}^d x \sqrt{\bar{g}} \, (\mathfrak{Q}^\dagger_{\mu\nu} Y) \, \theta \,, \tag{54}$$

upon neglecting boundary terms. By this it is clear that $\mathfrak{Q}^\dagger$ annihilates the gauge condition,

$$\mathfrak{Q}^{\dagger\mu\nu} \mathfrak{F}^\alpha{}_{\mu\nu} = 0 \,. \tag{55}$$

Also, $\mathfrak{Q}^\dagger$ annihilates $h^{\mathrm{TT}}$,

$$\mathfrak{Q}^{\dagger\mu\nu} h^{\mathrm{TT}}_{\mu\nu} = 0 \,. \tag{56}$$

Note that $\mathfrak{F}^\dagger = -\mathfrak{F}$ since it is a linear differential operator, so we will omit the dagger symbol for it.

Combining all properties, we see that in the calculation of the Gaussian integral

$$\int \mathcal{D}h \, e^{-\int d^d x \, \sqrt{\bar{g}} \, h_{\mu\nu} h^{\mu\nu}} , \tag{57}$$

all off-diagonal terms, that is those that mix the different modes, vanish. We also see immediately that the transverse traceless sector does not give rise to a Jacobian. The gauge vector integral reads

$$\int \mathcal{D}\zeta \, \mathcal{J}_\zeta \, e^{-\int d^d x \, \sqrt{\bar{g}} \, \frac{1}{2} \zeta^\mu \left[ -2\mathfrak{F}_\mu{}^{\alpha\beta} \mathfrak{F}^\nu{}_{\alpha\beta} \right] \zeta_\nu} , \tag{58}$$

so that the corresponding Jacobian is

$$\mathcal{J}_\zeta = \left( \det \left[ -2\mathfrak{F}_\mu{}^{\alpha\beta} \mathfrak{F}^\nu{}_{\alpha\beta} \right] \right)^{1/2} . \tag{59}$$

The choice of normalisation will be made clear below. In the gauge invariant scalar sector,

$$\int \mathcal{D}\theta \, \mathcal{J}_\theta \, e^{-\int d^d x \, \sqrt{\bar{g}} \, \theta \mathfrak{Q}^{\dagger\mu\nu} \mathfrak{Q}_{\mu\nu} \theta} , \tag{60}$$

so that the Jacobian reads

$$\mathcal{J}_\theta = \left( \det \mathfrak{Q}^{\dagger\mu\nu} \mathfrak{Q}_{\mu\nu} \right)^{1/2} . \tag{61}$$

**Avoiding the Jacobians**   Once again, we would like to avoid the introduction of these Jacobians. For that matter, we rescale the fields by

$$\zeta_\mu = \left[ \left( -2\mathfrak{F}_.{}^{\alpha\beta} \mathfrak{F}_{\alpha\beta} \right)^{-1/2} \right]^\nu_\mu \xi_\nu , \qquad \theta = \left( \mathfrak{Q}^{\dagger\mu\nu} \mathfrak{Q}_{\mu\nu} \right)^{-1/2} \sigma , \tag{62}$$

again assuming that all involved operators exist. The decomposition of the metric fluctuation into the set $(h^{\mathrm{TT}}, \xi, \sigma)$,

$$h_{\mu\nu} = h_{\mu\nu}^{\mathrm{TT}} + 2\mathfrak{F}^\alpha{}_{\mu\nu} \left[ \left( -2\mathfrak{F}_.{}^{\kappa\lambda} \mathfrak{F}_{\kappa\lambda} \right)^{-1/2} \right]^\beta_\alpha \xi_\beta + \mathfrak{Q}_{\mu\nu} \left( \mathfrak{Q}^{\dagger\alpha\beta} \mathfrak{Q}_{\alpha\beta} \right)^{-1/2} \sigma , \tag{63}$$

then gives rise to no Jacobians, and the decomposed fields have all the same mass dimension.

**Projectors I**   We can now construct the projectors onto each of the individual components. In doing so, we make use of the properties of the gauge fixing operator $\mathfrak{F}$ and of $\mathfrak{Q}$. Let us start with the gauge invariant scalar. Acting with $\mathfrak{Q}^\dagger$ on (63) gives

$$\mathfrak{Q}^{\dagger\mu\nu} h_{\mu\nu} = \left( \mathfrak{Q}^{\dagger\alpha\beta} \mathfrak{Q}_{\alpha\beta} \right)^{1/2} \sigma . \tag{64}$$

From this it is immediately clear that the projector onto this mode reads

$$\Pi^0{}_{\mu\nu}{}^{\kappa\lambda} = \mathfrak{Q}_{\mu\nu} \left( \mathfrak{Q}^{\dagger\alpha\beta} \mathfrak{Q}_{\alpha\beta} \right)^{-1} \mathfrak{Q}^{\dagger\kappa\lambda} . \tag{65}$$

In a similar fashion, acting with the gauge fixing operator onto $h$ gives

$$\mathfrak{F}_\alpha{}^{\mu\nu} h_{\mu\nu} = -\left[ \left( -2\mathfrak{F}_.{}^{\kappa\lambda} \mathfrak{F}_{\kappa\lambda} \right)^{1/2} \right]^\nu_\alpha \xi_\nu , \tag{66}$$

so that the gauge projector is

$$\Pi^1{}_{\mu\nu}{}^{\kappa\lambda} = -2\mathfrak{F}^\alpha{}_{\mu\nu} \left[ \left( -2\mathfrak{F}_.{}^{\tau\omega} \mathfrak{F}_{\tau\omega} \right)^{-1} \right]^\beta_\alpha \mathfrak{F}_\beta{}^{\kappa\lambda} . \tag{67}$$

Finally, we define the projector onto the TT mode by subtracting the two other projectors from the symmetric identity,

$$\Pi^2_{\mu\nu}{}^{\kappa\lambda} = \mathbb{1}_{\mu\nu}{}^{\kappa\lambda} - \Pi^1_{\mu\nu}{}^{\kappa\lambda} - \Pi^0_{\mu\nu}{}^{\kappa\lambda}. \tag{68}$$

The symmetric identity is defined as

$$\mathbb{1}_{\mu\nu}{}^{\kappa\lambda} = \delta_{(\mu}{}^{\kappa}\delta_{\nu)}{}^{\lambda}, \tag{69}$$

and maps symmetric rank two tensors to themselves. Inserting explicit expressions into these projectors seems to indicate that $\Pi^2$ depends on $\beta$. We will now rewrite everything to show that this is actually not the case.

**Rewriting the operators** In the above expressions, we have two different inverse operators, one constructed from the square of the gauge fixing operator,

$$-2\mathfrak{F}_\mu{}^{\alpha\beta}\mathfrak{F}^\nu{}_{\alpha\beta} = \bar{\Delta}\delta_\mu{}^\nu - \bar{D}^\nu\bar{D}_\mu + 2\frac{1-\beta^2}{d}\bar{D}_\mu\bar{D}^\nu, \tag{70}$$

whose explicit form clarifies the choice of prefactor, the other is $\mathcal{X}$ which appears in the operator $\mathfrak{Q}$,

$$\mathcal{X} = \left[ \bar{\Delta}\delta_{\cdot}^{\cdot} - \bar{D}^{\cdot}\bar{D}_{\cdot} + 2\frac{1+\beta}{d}\bar{D}_{\cdot}\bar{D}^{\cdot} \right]^{-1}. \tag{71}$$

The two operators agree for the gauge parameter choices $\beta = 0, -1$. It will be convenient to formally expand the operators in a Taylor series in $\beta$ around zero, and resum the full series once the inverse is calculated. The central operator then is

$$\Delta_{1\mu}{}^\nu = \bar{\Delta}\delta_\mu{}^\nu - \bar{D}^\nu\bar{D}_\mu + \frac{2}{d}\bar{D}_\mu\bar{D}^\nu. \tag{72}$$

Once again we assume that the inverse of $\Delta_1$ exists. Using a geometric series, we can write $\mathcal{X}$ as

$$\begin{aligned}
\mathcal{X}_\mu{}^\nu &= \sum_{l\geq 0}\left[\left(-\frac{2\beta}{d}\left[\Delta_1^{-1}\right]_{\cdot}^{\gamma}\bar{D}_\gamma\bar{D}^{\cdot}\right)^l\right]_\mu{}^\alpha \left[\Delta_1^{-1}\right]_\alpha{}^\nu \\
&= \left[\Delta_1^{-1}\right]_\mu{}^\nu - \frac{2\beta}{d}\left[\Delta_1^{-1}\right]_\mu{}^\alpha \bar{D}_\alpha \sum_{l\geq 0}\left(-\frac{2\beta}{d}\bar{D}\cdot\Delta_1^{-1}\cdot\bar{D}\right)^l \bar{D}^\gamma\left[\Delta_1^{-1}\right]_\gamma{}^\nu \\
&= \left[\Delta_1^{-1}\right]_\mu{}^\nu - \frac{2\beta}{d}\left[\Delta_1^{-1}\right]_\mu{}^\alpha \bar{D}_\alpha \frac{1}{1+\frac{2\beta}{d}\bar{D}\cdot\Delta_1^{-1}\cdot\bar{D}}\bar{D}^\gamma\left[\Delta_1^{-1}\right]_\gamma{}^\nu \\
&\equiv \left[\Delta_1^{-1}\right]_\mu{}^\nu - \frac{2\beta}{d}\left[\Delta_1^{-1}\right]_\mu{}^\alpha \bar{D}_\alpha \frac{1}{1+\frac{2\beta}{d}\mathcal{N}}\bar{D}^\gamma\left[\Delta_1^{-1}\right]_\gamma{}^\nu.
\end{aligned} \tag{73}$$

From the first to the second line, we rewrote the terms in the sum into a form of another geometric series, which is performed in the next step. We also defined the scalar operator

$$\mathcal{N} = \bar{D}^\mu\left(\Delta_1^{-1}\right)_\mu{}^\nu\bar{D}_\nu. \tag{74}$$

The inverse of the squared gauge fixing operator (70) can evidently be obtained from that result by the replacement $\beta \to -\beta^2$, so that

$$\left[\left(-2\mathfrak{F}_{\cdot}^{\tau\omega}\mathfrak{F}^{\cdot}{}_{\tau\omega}\right)^{-1}\right]_\mu{}^\nu = \left[\Delta_1^{-1}\right]_\mu{}^\nu + \frac{2\beta^2}{d}\left[\Delta_1^{-1}\right]_\mu{}^\alpha \bar{D}_\alpha \frac{1}{1-\frac{2\beta^2}{d}\mathcal{N}}\bar{D}^\gamma\left[\Delta_1^{-1}\right]_\gamma{}^\nu. \tag{75}$$

Before we go back to the explicit form of the projectors, we can re-express the operator $\mathfrak{Q}$ as

$$\mathfrak{Q}_{\mu\nu} = \bar{g}_{\mu\nu} - 2\beta \, \bar{D}_{(\mu} \left[ \Delta_1^{-1} \right]_{\nu)}{}^{\alpha} \bar{D}_{\alpha} \frac{1}{1 + \frac{2\beta}{d}\mathcal{N}} \,. \tag{76}$$

This also entails the compact form of the expression

$$\mathfrak{Q}^{\dagger\mu\nu}\mathfrak{Q}_{\mu\nu} = d \, \frac{1 - \frac{2\beta^2}{d}\mathcal{N}}{(1 + \frac{2\beta}{d}\mathcal{N})^2} \,. \tag{77}$$

**Projectors II** We will now present the explicit form of all projectors. The scalar projector reads

$$\Pi^0{}_{\mu\nu}{}^{\kappa\lambda} = \left[ \bar{g}_{\mu\nu} - \bar{D}_{(\mu} \left[ \Delta_1^{-1} \right]_{\nu)}{}^{\alpha} \bar{D}_{\alpha} \frac{2\beta}{1 + \frac{2\beta}{d}\mathcal{N}} \right] \times$$
$$\frac{\left( 1 + \frac{2\beta}{d}\mathcal{N} \right)^2}{d \left( 1 - \frac{2\beta^2}{d}\mathcal{N} \right)} \left[ \bar{g}^{\kappa\lambda} - \frac{2\beta}{1 + \frac{2\beta}{d}\mathcal{N}} \bar{D}^{\gamma} \left[ \Delta_1^{-1} \right]_{\gamma}{}^{(\kappa} \bar{D}^{\lambda)} \right]. \tag{78}$$

For the gauge projector we find, after a short calculation,

$$\Pi^1{}_{\mu\nu}{}^{\kappa\lambda} = -2\bar{D}_{(\mu} \left[ \Delta_1^{-1} \right]_{\nu)}{}^{(\kappa} \bar{D}^{\lambda)} - \frac{4\beta^2}{d} \bar{D}_{(\mu} \left[ \Delta_1^{-1} \right]_{\nu)}{}^{\alpha} \bar{D}_{\alpha} \frac{1}{1 - \frac{2\beta^2}{d}\mathcal{N}} \bar{D}^{\gamma} \left[ \Delta_1^{-1} \right]_{\gamma}{}^{(\kappa} \bar{D}^{\lambda)}$$
$$+ 2\frac{1+\beta}{d} \left[ \bar{g}_{\mu\nu} \frac{1}{1 - \frac{2\beta^2}{d}\mathcal{N}} \bar{D}^{\alpha} \left[ \Delta_1^{-1} \right]_{\alpha}{}^{(\kappa} \bar{D}^{\lambda)} + \bar{D}_{(\mu} \left[ \Delta_1^{-1} \right]_{\nu)}{}^{\alpha} \bar{D}_{\alpha} \frac{1}{1 - \frac{2\beta^2}{d}\mathcal{N}} \bar{g}^{\kappa\lambda} \right] \tag{79}$$
$$- 2 \left( \frac{1+\beta}{d} \right)^2 \bar{g}_{\mu\nu} \bar{g}^{\kappa\lambda} \frac{\mathcal{N}}{1 - \frac{2\beta^2}{d}\mathcal{N}} \,.$$

Combining the two into the TT projector, we get

$$\Pi^2{}_{\mu\nu}{}^{\kappa\lambda} = \Pi^{\mathrm{TL}}{}_{\mu\nu}{}^{\alpha\beta} \left[ \mathbb{1}_{\alpha\beta}{}^{\gamma\delta} + 2\bar{D}_{(\alpha} \left[ \Delta_1^{-1} \right]_{\beta)}{}^{(\gamma} \bar{D}^{\delta)} \right] \Pi^{\mathrm{TL}}{}_{\gamma\delta}{}^{\kappa\lambda} \,. \tag{80}$$

Here we used the traceless projector to bring the expression into a compact form,

$$\Pi^{\mathrm{TL}}{}_{\mu\nu}{}^{\kappa\lambda} = \mathbb{1}_{\mu\nu}{}^{\kappa\lambda} - \frac{1}{d} \bar{g}_{\mu\nu} \bar{g}^{\kappa\lambda} \equiv \mathbb{1}_{\mu\nu}{}^{\kappa\lambda} - \Pi^{\mathrm{Tr}}{}_{\mu\nu}{}^{\kappa\lambda} \,, \tag{81}$$

where in the second equation we also introduced the trace projector $\Pi^{\mathrm{Tr}}$. As promised above, the TT-projector is indeed independent of the gauge parameter $\beta$.

**Natural operator** An obvious question is whether we can define a natural operator for the graviton. This would be a local Laplace-type operator which commutes with the projectors. As it turns out, such an operator does not exist. One can show this in the following way. Assume that there is an operator $\mathbf{\Delta}_2$ which commutes with the spin two projector (80). In that case, we would have that

$$\bar{D}^{\mu}\mathbf{\Delta}_{2\mu\nu}{}^{\rho\sigma}h^{\mathrm{TT}}_{\rho\sigma} = \bar{D}^{\mu}\Pi^2{}_{\mu\nu}{}^{\kappa\lambda}\mathbf{\Delta}_{2\kappa\lambda}{}^{\rho\sigma}h^{\mathrm{TT}}_{\rho\sigma} = 0 \,, \tag{82}$$

since the projector is transverse. We can easily write down the most general form that this local operator can take,

$$\mathbf{\Delta}_{2\mu\nu}{}^{\rho\sigma} = \left( \bar{\Delta} + c_1\bar{R} \right) \Pi^{\mathrm{TL}}{}_{\mu\nu}{}^{\kappa\lambda} + c_2 \, \Pi^{\mathrm{TL}}{}_{\mu\nu}{}^{\alpha\beta} \bar{R}_{\alpha}{}^{\gamma}\delta_{\beta}{}^{\delta} \Pi^{\mathrm{TL}}{}_{\gamma\delta}{}^{\kappa\lambda} + c_3 \, \bar{C}_{(\mu}{}^{\rho}{}_{\nu)}{}^{\sigma} \,. \tag{83}$$

Here $\bar{C}$ is the background Weyl tensor, see (108) below, and the $c_i$ are numerical coefficients. All other potential tensor structures vanish when they act on $h^{\mathrm{TT}}$. Inserting this ansatz into (82), one finds that there is no choice of $c_i$ to make this equation true. One can find a non-local solution to (82), but due to the inherent difficulties in handling such operators, we will avoid that path in this work, and rather look for alternatives for the operator that we want to regularise.

## 3.4 Regularisation and decoupling in gravity

Having discussed the decomposition of the graviton into gauge variant and invariant modes, but not having found a natural operator, we now have to construct a regularisation scheme by other means. Let us first discuss the gauge variant vector mode. By means of the decoupling theorem [12], the vector mode decouples from the gauge invariant modes completely in the Landau gauge limit, which implements the gauge fixing condition strictly. This means that the functional trace (1) splits into the gauge invariant sector which involves the correlation functions derived from the given action, and a simple vector trace of the form (38) with the operator (70). At the same time, the trace over the Faddeev-Popov ghosts is the same trace but with the operator (34). As noted earlier, the two operators agree if either $\beta = -1$ or $\beta = 0$. These gauge choice thus implement an exact partial cancellation of these traces. The cancellation is only partial due to the extra factor of two for the ghosts. The regularisation of these two vector fields then follows the discussion in subsection 3.2.

Now we will discuss the gauge invariant part. From the explicit form of the spin zero projector (78), we see that the gauge choice $\beta = -1$ leaves the gauge invariant scalar non-local. We would like to avoid such non-localities, so we will settle for $\beta = 0$ as our preferred gauge choice. In this case, the gauge invariant scalar mode is just the trace of the fluctuation field $h$.

To finally fix the regularisation, we take inspiration from the Abelian case and consider the two-point correlation function of the simplest gravitational action - the Einstein-Hilbert action (without cosmological constant):

$$S^{\mathrm{EH}} \propto \frac{1}{G_N} \int \mathrm{d}^4 x \sqrt{g} R. \tag{84}$$

We then decompose the field via (63) where we can neglect the gauge variant vector since it decouples. In $d = 4$, the two-point function turns out to be diagonal, and up to overall prefactors containing $G_N$, it has the two parts

$$h^{\mathrm{TT}}_{\mu\nu} \triangle_2^{\mu\nu\rho\sigma} h^{\mathrm{TT}}_{\rho\sigma} \equiv h^{\mathrm{TT}}_{\mu\nu}\left[\left(\bar{\triangle} + \frac{2}{3}\bar{R}\right)\Pi^{\mathrm{TL}\mu\nu\rho\sigma} - 2\bar{C}^{(\mu|\rho|\nu)\sigma}\right]h^{\mathrm{TT}}_{\rho\sigma}, \qquad h\bar{\triangle}h. \tag{85}$$

Here $h = h_\mu{}^\mu$ is the trace of $h_{\mu\nu}$, which is the field $\sigma$ for $\beta = 0$. We thus propose a regularisation involving $\triangle_2$ and $\triangle$, with traceless and trace projectors, respectively:

$$\mathfrak{R}^{\mathrm{TT}}_k(\triangle_2) \propto \frac{k^2}{G_N}\Pi^{\mathrm{TL}} \cdot \mathcal{R}^{\mathrm{TT}}_k(\triangle_2/k^2) \cdot \Pi^{\mathrm{TL}},$$

$$\mathfrak{R}^{\mathrm{Tr}}_k(\bar{\triangle}) \propto \frac{k^2}{G_N}\Pi^{\mathrm{Tr}} \mathcal{R}^{\mathrm{Tr}}_k(\bar{\triangle}/k^2). \tag{86}$$

The numerical prefactors are fixed uniquely by requiring that the regularised version of (85) reads

$$h^{\mathrm{TT}}_{\mu\nu}\left[\triangle_2 + k^2 \mathcal{R}^{\mathrm{TT}}_k(\triangle_2/k^2)\right]^{\mu\nu\rho\sigma} h^{\mathrm{TT}}_{\rho\sigma}, \qquad h\left[\bar{\triangle} + k^2 \mathcal{R}^{\mathrm{Tr}}_k(\bar{\triangle}/k^2)\right]h. \tag{87}$$

Let us explain why we do not have to use the transverse traceless projectors for the regulator. First, the regulator clearly does not add a regularisation to the trace mode by construction,

since we chose $\beta = 0$. Second, the regulator has some overlap with the gauge variant vector mode. However, since we implement the strict Landau gauge limit, the contribution of this regulator to the vector mode drops out of any final trace. It is thus unnecessary to employ the spin two projector, and we can rather use the much easier traceless projector. Note that this regulator also fulfils our mode count requirement.

Let us note that in dimensions other than four, the two-point function obtained from the Einstein-Hilbert action is not diagonal - there are off-diagonal terms of the form

$$h_{\mu\nu}^{\mathrm{TT}} \bar{R}^{\mu\nu} h. \tag{88}$$

We will not discuss the regularisation with such off-diagonal terms here, since we are exclusively interested in the physical case $d = 4$ in this work.

**(Not) Using the decomposition**    Having discussed the regularisation, we will briefly clarify why using a decomposition in practice does not necessarily simplify computations except in special cases. The central reason is that the inverse of a projected operator (in the projected subspace) is in general not the projected inverse of the unprojected operator. As a concrete example (neglecting the regularisation), assume that we would want to invert a plain Laplacian in the transverse sector of a vector field. That is, we are looking for the inverse

$$\left[ \Pi^{\mathrm{T}} \cdot \Delta \cdot \Pi^{\mathrm{T}} \right]^{-1}, \tag{89}$$

where the inversion is to be understood in the transverse subspace. To compute this, we can complete the Laplacian to the natural vector operator $\Delta_A$ and invert the operator via a geometric series:

$$\begin{aligned}
\left[ \Pi^{\mathrm{T}} \cdot \Delta \cdot \Pi^{\mathrm{T}} \right]^{-1} &= \left[ \Pi^{\mathrm{T}} \cdot (\Delta_A - \mathrm{Ric}) \cdot \Pi^{\mathrm{T}} \right]^{-1} \\
&= \left[ \Pi^{\mathrm{T}} \cdot \Delta_A \cdot \left( \Pi^{\mathrm{T}} - \frac{1}{\Delta_A} \cdot \Pi^{\mathrm{T}} \cdot \mathrm{Ric} \cdot \Pi^{\mathrm{T}} \right) \right]^{-1} \\
&= \sum_{n \geq 0} \Pi^{\mathrm{T}} \cdot \left( \frac{1}{\Delta_A} \cdot \Pi^{\mathrm{T}} \cdot \mathrm{Ric} \cdot \Pi^{\mathrm{T}} \right)^n \cdot \frac{1}{\Delta_A} \cdot \Pi^{\mathrm{T}}.
\end{aligned} \tag{90}$$

Here Ric indicates the Ricci tensor. By contrast, the projected inverse reads

$$\begin{aligned}
\Pi^{\mathrm{T}} \cdot \frac{1}{\Delta} \Pi^{\mathrm{T}} &= \Pi^{\mathrm{T}} \cdot \frac{1}{\Delta_A - \mathrm{Ric}} \cdot \Pi^{\mathrm{T}} \\
&= \sum_{n \geq 0} \Pi^{\mathrm{T}} \cdot \left( \frac{1}{\Delta_A} \cdot \mathrm{Ric} \right)^n \cdot \frac{1}{\Delta_A} \cdot \Pi^{\mathrm{T}}.
\end{aligned} \tag{91}$$

These two expressions do not agree for an arbitrary manifold. In general, the two expressions only agree if the operator commutes with the projector. This once again highlights the central role of a natural operator.

Coming back to gravity, in the previous subsection we found that there is no local natural operator for the transverse traceless part of the graviton. Rather, the Einstein-Hilbert action suggests to consider the spin two operator $\triangle_2$, which does not commute with the spin two projector $\Pi^2$. As a consequence,

$$\left[ \Pi^2 \cdot \triangle_2 \cdot \Pi^2 \right]^{-1} \neq \Pi^2 \cdot \triangle_2^{-1} \cdot \Pi^2. \tag{92}$$

The inversion on the left hand side is to be understood on the space of transverse traceless tensors. This means that even if we use the decomposition, the computation of the transverse traceless propagator is complicated by the presence of the projectors.

## 4 Inversion of the graviton two-point function

To compute the non-perturbative renormalisation group flow with the FRG, we have to compute the regularised propagator, which is the inverse of the regularised two-point function. In this section we briefly illustrate how to perform this inversion for the graviton in a derivative expansion, without specifying a particular background metric. The idea is based on the observation that if we were to compute the propagator in flat spacetime, that is to zeroth order in the derivative expansion, we can simply go to momentum space and perform the inversion with standard techniques. Concretely, both the flat two-point function and the flat propagator have the form

$$
\begin{aligned}
X_{\mu\nu}{}^{\rho\sigma} = {} & X_1(p^2) \mathbb{1}_{\mu\nu}{}^{\rho\sigma} + X_2(p^2) \Pi^{\mathrm{Tr}}{}_{\mu\nu}{}^{\rho\sigma} + X_3(p^2) \delta^{(\rho}_{(\mu} p_{\nu)} p^{\sigma)} \\
& + \frac{1}{2} X_4(p^2) \big( g_{\mu\nu} p^\rho p^\sigma + p_\mu p_\nu g^{\rho\sigma} \big) + X_5(p^2) p_\mu p_\nu p^\rho p^\sigma \,.
\end{aligned}
\tag{93}
$$

Here, $p_\mu$ is the momentum vector. The determination of the flat part of the propagator from an arbitrary flat two-point function has been carried out in full detail *e.g.* in [41]. The key observation is that any difference between the flat propagator and the full propagator is, by definition, at least linear in curvature. We can thus define the covariantised flat propagator

$$
\begin{aligned}
G_{0\mu\nu}{}^{\rho\sigma} = {} & g_1(\bar\Delta) \mathbb{1}_{\mu\nu}{}^{\rho\sigma} + g_2(\bar\Delta) \Pi^{\mathrm{Tr}}{}_{\mu\nu}{}^{\rho\sigma} + \delta^{(\rho}_{(\mu} \bar{D}_{\nu)} \bar{D}^{\sigma)} g_3(\bar\Delta) \\
& + \frac{1}{2} \big( g_{\mu\nu} \bar{D}^{(\rho} \bar{D}^{\sigma)} + \bar{D}_{(\mu} \bar{D}_{\nu)} g^{\rho\sigma} \big) g_4(\bar\Delta) + \bar{D}_{(\mu} \bar{D}_{\nu)} \bar{D}^{(\rho} \bar{D}^{\sigma)} g_5(\bar\Delta).
\end{aligned}
\tag{94}
$$

Crucially, we can *choose* where we put the scalar propagator functions $g_i$, since any different choice only differs by terms at least linear in curvature. The functions $g_i$ can be computed in flat space. To obtain the full propagator, we notice that the product of the inverse of the full propagator $G$ and $G_0$ is

$$
G^{-1} \cdot G_0 = \mathbb{1} + \mathfrak{X} \,,
\tag{95}
$$

where $\mathfrak{X}$ is, by definition, of linear and higher order in curvature. Crucially, since the inverse full propagator is just the regularised two-point function, which is known for a given action, $\mathfrak{X}$ can be *computed* to the necessary order. Knowing $\mathfrak{X}$, we then can calculate the full propagator via

$$
G = G_0 \left[ \mathbb{1} + \mathfrak{X} \right]^{-1} = \sum_{l \geq 0} G_0 \, \mathfrak{X}^l \,,
\tag{96}
$$

where we suppressed the indices. For a fixed order of the derivative expansion, only finitely many terms of this sum contribute to the full propagator.

Let us mention that one can of course also choose a different operator ordering. In complete analogy to the above, we can define a tensor $\mathfrak{Y}$ via

$$
G_0 G^{-1} = \mathbb{1} + \mathfrak{Y} \,,
\tag{97}
$$

so that the full propagator reads

$$
G = \left[ \mathbb{1} + \mathfrak{Y} \right]^{-1} G_0 \,.
\tag{98}
$$

In general the tensors $\mathfrak{X}$ and $\mathfrak{Y}$ do not agree. Which of the two orderings is more efficient is hard to predict generally, and has to be tested in practice.

The algorithm also works on more general backgrounds, for which one can derive the exact propagator. A prime example is the sphere - the Ricci scalar is finite and covariantly constant, so that (94) holds if we let the propagator functions also depend on $\bar{R}$. This has been used in the context of affine gravity in [155].

# 5 Commutator rules

The complexity of computations beyond actions linear in the curvature increases extremely quickly. It is thus advantageous to employ tensor algebra packages to perform the necessary calculations to achieve reliable results. A key ingredient for a reliable code is the generic implementation of simplification rules like commutators to a given order. In this section, we will derive such recursive formulas for the commutator of a function of the Laplacian with either a curvature tensor or a covariant derivative. Our focus lies on formulas applicable in a finite order derivative expansion - an extension to the curvature expansion will be presented elsewhere.

Before we specify a detailed commutator, let us consider the following general case. Let $f$ be a suitable function, $X$ is an arbitrary operator and $Y$ is a tensor of arbitrary rank. We are interested in a formula of the form

$$f(\bar{\Delta})XY = Xf(\bar{\Delta})Y + \dots, \tag{99}$$

where we want to find an explicit expression of the term indicated by the dots on the right-hand side of the equation. To derive the expression, we will formally use an inverse Laplace transform and the Baker-Campbell-Hausdorff formula,

$$
\begin{aligned}
f(\bar{\Delta})XY &= \int_0^\infty \mathrm{d}s\, \tilde{f}(s) e^{-s\bar{\Delta}} XY \\
&= Xf(\bar{\Delta})Y + \int_0^\infty \mathrm{d}s\, \tilde{f}(s) \sum_{l\geq 1} \frac{(-s)^l}{l!} \left[\bar{\Delta}, X\right]_l e^{-s\bar{\Delta}} Y \\
&= Xf(\bar{\Delta})Y + \sum_{l\geq 1} \frac{1}{l!} \left[\bar{\Delta}, X\right]_l f^{(l)}(\bar{\Delta}) Y.
\end{aligned}
\tag{100}
$$

In this equation, we use the multicommutator

$$[A,B]_n = [A, [A,B]_{n-1}], \qquad [A,B]_1 \equiv [A,B] = AB - BA. \tag{101}$$

The reason why (100) is useful in a derivative expansion is that the multicommutators increase the order of the expression by at least one. In that way, in a finite order computation, only finitely many terms in this sum contribute.

With the help of the formulas that we prove in appendix A, we can rewrite (100) into the form

$$f(\bar{\Delta})XY = Xf(\bar{\Delta})Y + \sum_{l\geq 1} \frac{1}{l!} \sum_{k=0}^{l-1} (-1)^{l-1-k} \binom{l-1}{k} \bar{\Delta}^k \left[\bar{\Delta}, X\right] \bar{\Delta}^{l-k-1} f^{(l)}(\bar{\Delta}) Y. \tag{102}$$

The usefulness of this formula lies in the fact that it only involves the simple commutator, which is straightforward to implement.

Let us now specify the two cases of commutators that are needed in the derivative expansion. The first case is whenever $X$ is a multiplication operator. The relevant example is that of a curvature tensor, potentially with a number of derivatives acting on it. In that case, we have

$$\left[\bar{\Delta}, X\right] = \left(\bar{\Delta}X\right) - 2\left(\bar{D}^\mu X\right)\bar{D}_\mu. \tag{103}$$

This can be inserted into (102) and produces

$$
\begin{aligned}
f(\bar{\Delta})XY &= Xf(\bar{\Delta})Y \\
&+ \sum_{l\geq 1} \frac{1}{l!} \sum_{k=0}^{l-1} (-1)^{l-1-k} \binom{l-1}{k} \bar{\Delta}^k \left\{\left(\bar{\Delta}X\right) - 2\left(\bar{D}^\mu X\right)\bar{D}_\mu\right\} \bar{\Delta}^{l-k-1} f^{(l)}(\bar{\Delta}) Y.
\end{aligned}
\tag{104}
$$

Since the commutator increases the order of the expression by at least one derivative, the recursive application of (102) produces only finitely many terms for a fixed order of the derivative expansion.

The second case is when $X = \bar{D}_\mu$ is the covariant derivative. In that case,

$$\left[\bar{\Delta}, \bar{D}_\mu\right] = -\bar{D}^\alpha\left[\bar{D}_\alpha, \bar{D}_\mu\right] - \left[\bar{D}^\alpha, \bar{D}_\mu\right]\bar{D}_\alpha. \tag{105}$$

The commutator of two covariant derivatives is related to the Riemann tensor via

$$\left[\bar{D}_\mu, \bar{D}_\nu\right]Y_{\alpha_1\ldots\alpha_n} = \sum_{l=1}^{n} \bar{R}_{\mu\nu\alpha_l}{}^\beta Y_{\alpha_1\ldots\alpha_{l-1}\beta\alpha_{l+1}\ldots\alpha_n}. \tag{106}$$

We thus find

$$
\begin{aligned}
f(\bar{\Delta})\bar{D}_\mu Y_{\alpha_1\ldots\alpha_n} = {}& \bar{D}_\mu f(\bar{\Delta})Y_{\alpha_1\ldots\alpha_n} - \sum_{l\geq 1}\frac{1}{l!}\sum_{k=0}^{l-1}(-1)^{l-1-k}\binom{l-1}{k}\bar{\Delta}^k \times \\
& \left[\bar{R}_{\alpha\mu}\bar{D}^\alpha Y_{\alpha_1\ldots\alpha_n} + \bar{D}^\alpha\sum_{l=1}^{n}\bar{R}_{\alpha\mu\alpha_l}{}^\beta\bar{\Delta}^{l-k-1}f^{(l)}(\bar{\Delta})Y_{\alpha_1\ldots\alpha_{l-1}\beta\alpha_{l+1}\ldots\alpha_n}\right. \\
& \left. + \sum_{l=1}^{n}\bar{R}_{\alpha\mu\alpha_l}{}^\beta\bar{D}^\alpha\bar{\Delta}^{l-k-1}f^{(l)}(\bar{\Delta})Y_{\alpha_1\ldots\alpha_{l-1}\beta\alpha_{l+1}\ldots\alpha_n}\right].
\end{aligned}
\tag{107}
$$

This time, the commutator increases the order of the expression by at least two units, so that once again only finitely many terms contribute to any fixed order computation. Repeatedly applying the formulas (104) and (107) then gives the commutator of a function of the Laplace operator to the needed order.

# 6 Simplification of tensor expressions of maximal order

Before we finally discuss the application of our framework, we shall point out a way to simplify the calculation of truncated RG flows significantly. This simplification concerns operators which are already of the derivative order that one truncates at, and before such operators are traced. The key observation is that eventually these terms will be contracted with metrics only, and by assumption no higher order terms arise. Because of this, we can replace these operators by combinations of metrics and *scalar* curvature invariants that respect the symmetries of the term.

To make this concrete, let us first discuss the Einstein-Hilbert case, that is we truncate at the second order in derivatives. At this order, the most convenient way to see the simplification is to transition to the traceless basis, that is all occurrences of the Riemann tensor are replaced by the Weyl tensor $\bar{C}$, the Ricci tensor and the Ricci scalar via

$$\bar{R}_{\mu\nu}{}^{\rho\sigma} = \bar{C}_{\mu\nu}{}^{\rho\sigma} + \frac{4}{d-2}\delta_{[\mu}{}^{[\rho}\bar{R}_{\nu]}{}^{\sigma]} - \frac{2}{(d-2)(d-1)}\bar{R}\,\delta_\mu{}^{[\rho}\delta_\nu{}^{\sigma]}, \tag{108}$$

and then all Ricci tensors are replaced by traceless Ricci tensors $\bar{S}$ and Ricci scalars via

$$\bar{R}_{\mu\nu} = \bar{S}_{\mu\nu} + \frac{1}{d}\bar{g}_{\mu\nu}\bar{R}. \tag{109}$$

Now since eventually all these tensors must be contracted with metrics only, it is immediately clear that we can drop the terms with Weyl and traceless Ricci tensors since their traces vanish.

In other words, terms linear in $\bar{C}$ and $\bar{S}$ can only contribute to quartic and higher orders in derivatives, and we can set

$$\bar{C}_{\mu\nu\rho\sigma} \mapsto 0,$$
$$\bar{S}_{\mu\nu} \mapsto 0. \tag{110}$$

Indeed this has been used implicitly in much of the literature on Asymptotic Safety, however with a different view, namely that a special background was chosen. Here we see that the procedure is indeed general and not related to a specific choice of background.

At quartic order in derivatives the structure is slightly more complicated. In this case we can replace terms quartic in the curvature, but not those that are linear. Once again it is useful to employ the traceless basis. Since there are only three scalar curvature monomials at this order, in this basis only three combinations of curvatures do not vanish. In particular we find directly that

$$\bar{C}_{\mu\nu\rho\sigma}\bar{S}_{\tau\omega} \mapsto 0,$$
$$\bar{C}_{\mu\nu\rho\sigma}\bar{R} \mapsto 0,$$
$$\bar{S}_{\mu\nu}\bar{R} \mapsto 0, \tag{111}$$

since all complete contractions of these terms with metrics vanish. On the other hand, we find that

$$\bar{C}_{\mu\nu\rho\sigma}\bar{C}_{\tau\omega\kappa\lambda} \mapsto \mathcal{T}_{\mu\nu\rho\sigma\tau\omega\kappa\lambda}\bar{C}^{\alpha\beta\gamma\delta}\bar{C}_{\alpha\beta\gamma\delta},$$
$$\bar{S}_{\mu\nu}\bar{S}_{\rho\sigma} \mapsto \frac{2}{(d-1)(d+2)}\Pi^{\mathrm{TL}}_{\mu\nu\rho\sigma}\bar{S}^{\alpha\beta}\bar{S}_{\alpha\beta}. \tag{112}$$

Here $\mathcal{T}$ is a rank 8 tensor constructed from the metric alone, which is too long to be displayed here. These equations can be derived by making the most general ansatz of the appropriate number of metrics, imposing the relevant symmetries, and finally computing some particular contractions to fix any remaining free coefficients.

Note that both at quadratic and quartic order, if we neglect boundary terms (which we do in this work), all curvature tensors can be assumed to be covariantly constant. Only at sextic and higher orders, monomials with covariant derivatives appear.

Clearly one can also formulate similar equations in a Riemann basis, however the traceless basis disentangles the invariants maximally. The generalisation to higher orders is also straightforward, although increasingly lengthy. Nevertheless it is also clear that by using these relations the computational complexity can be decreased considerably, since a lot fewer tensor structures arise at any intermediate step of the calculation.

# 7 Application to quartic order

We will now apply the machinery introduced in the preceding sections to quantum gravity at the quartic order in the derivative expansion. This entails a theory space with a total of five coupling constants. This section contains a discussion of the action, the flow equations, the fixed point search strategy, the actual fixed point structure of the theory, and a discussion of the topological term.

## 7.1 Action

To fix our conventions, we will first discuss the ansatz for the effective average action. Concretely, this ansatz reads

$$\Gamma_k = \frac{1}{16\pi G_N} \int \mathrm{d}^4 x \sqrt{g} \left[ -R + 2\Lambda + \frac{1}{2}G_{C^2}C^{\mu\nu\rho\sigma}C_{\mu\nu\rho\sigma} - \frac{1}{6}G_{R^2}R^2 + G_{\mathfrak{E}}\mathfrak{E} \right]. \tag{113}$$

In this, $G_N$ is the Newton's constant, $\Lambda$ is the cosmological constant, and $G_{C^2}$, $G_{R^2}$ and $G_{\mathfrak{E}}$ are the quartic couplings. All these couplings depend on the FRG scale $k$, which for better readability we do not indicate explicitly. Moreover, $\mathfrak{E}$ stands for the integrand of the four-dimensional Euler characteristic,

$$\mathfrak{E} = R^{\mu\nu\rho\sigma}R_{\mu\nu\rho\sigma} - 4R^{\mu\nu}R_{\mu\nu} + R^2\,. \tag{114}$$

For the discussion of fixed points, we introduce dimensionless coupling constants by a rescaling with the appropriate power of $k$, so that

$$g = G_N k^2\,,\quad \lambda = \Lambda k^{-2}\,,\quad g_{C^2} = G_{C^2}k^2\,,\quad g_{R^2} = G_{R^2}k^2\,,\quad g_{\mathfrak{E}} = G_{\mathfrak{E}}k^2\,. \tag{115}$$

We will also use an overdot to indicate the derivative with respect to the RG time $t$, e.g.,

$$\dot{g} = \partial_t g\,. \tag{116}$$

The action (113) is amended by a gauge fixing term of the form

$$\Gamma_{\mathrm{gf}} = \frac{1}{32\pi G_N \alpha} \int \mathrm{d}^4 x \, \sqrt{\bar{g}} \, \bar{g}^{\alpha\beta}\Big(\mathfrak{F}_\alpha{}^{\mu\nu}h_{\mu\nu}\Big)\Big(\mathfrak{F}_\beta{}^{\rho\sigma}h_{\rho\sigma}\Big), \tag{117}$$

and a corresponding Faddeev-Popov ghost term

$$\Gamma_{\mathrm{c}} = \frac{1}{G_N} \int \mathrm{d}^4 x \, \sqrt{\bar{g}} \, \bar{c}^\mu \, \mathfrak{F}_\mu{}^{\alpha\beta} D_\alpha c_\beta\,. \tag{118}$$

Note that we deviate from standard convention by introducing the coupling $G_N$ also in the ghost action. This is necessary for an exact cancellation of traces as discussed in subsection 3.4. The gauge parameter $\alpha$ will be sent to zero to implement the Landau limit.

The final ingredient to specify is the regulator. Since this has been discussed in detail in section 3, we will not repeat it here.

To derive the flow equations, we have used the tensor algebra package suite *xAct* [156–160] together with a minimal extension[4] of [161] to parallelise the code.

## 7.2 Flow equations

In this section we present the flow equations for the dimensionless couplings of our system. For convenience, we introduce the dimensionless propagators

$$\begin{aligned}
\mathcal{G}^{\mathrm{TT}}(z) &= \frac{1}{z + g_{C^2}z^2 + \mathcal{R}_k^{\mathrm{TT}}(z) - 2\lambda}\,,\\
\mathcal{G}^{\mathrm{Tr}}(z) &= \frac{1}{z + g_{R^2}z^2 + \mathcal{R}_k^{\mathrm{Tr}}(z) - \frac{4}{3}\lambda}\,,\\
\mathcal{G}^{\mathrm{c}}(z) &= \frac{1}{z + \mathcal{R}^{\mathrm{c}}(z)}\,.
\end{aligned} \tag{119}$$

We also introduce the notation

$$\mathring{\mathfrak{R}}_k(z) = \left(4 - \frac{\dot{g}}{g}\right)\mathcal{R}_k(z) - 2z\,\mathcal{R}_k'(z), \tag{120}$$

for all regulator shape functions. It is related to the scale derivative of the regulator with some factors pulled out for convenience. The additional dependence on $\dot{g}$ comes from the fact that

---

[4]The extension consists of loading the package *xTras* in parallel to have access to the command *CollectTensors* on all kernels.

the regulator tensor comes with a prefactor of $1/g$, see (86), on which the scale derivative acts non-trivially. The additional factor of $1/g$ is then cancelled by a factor of $g$ coming from the propagator. This cancellation is already taken into account in the above notations.

For the cosmological constant, we find

$$\dot{\lambda} = \left(-4 + \frac{\dot{g}}{g}\right)\lambda + \frac{1}{2}\frac{8\pi g}{16\pi^2}\int_0^\infty \mathrm{d}z\, z\left[5\mathcal{G}^{\mathrm{TT}}(z)\mathring{\mathfrak{R}}_k^{\mathrm{TT}}(z) + \mathcal{G}^{\mathrm{Tr}}(z)\mathring{\mathfrak{R}}_k^{\mathrm{Tr}}(z) - 4\mathcal{G}^{\mathrm{c}}(z)\mathring{\mathfrak{R}}_k^{\mathrm{c}}(z)\right]. \quad (121)$$

We did not simplify the prefactor of the integral to illustrate where the individual factors come from: the factor $8\pi g$ comes from the left-hand side, the factor of a half comes from the flow equation itself, and the $1/(16\pi^2)$ comes from the heat kernel. We can also see how the mode count is satisfied: we have a prefactor of 5 for the transverse traceless sector, a prefactor of 1 for the trace sector, and a prefactor of $-4$ for the combination of gauge variant vector $(+4)$ and ghosts $(-8)$. For identical regulators and at vanishing cosmological constant and higher order couplings, this gives a total of 2 modes that contribute to the flow of the cosmological constant, which is the correct number of physical polarisations of the graviton in $d = 4$.

The flow of the dimensionless Newton's coupling reads

$$\dot{g} = 2g + \frac{1}{2}\frac{16\pi g^2}{16\pi^2}\int_0^\infty \mathrm{d}z\, \frac{1}{6}\left[\left\{-25 + 5\left(5g_{C^2} - 2g_{R^2}\right)z^2\mathcal{G}^{\mathrm{TT}}(z)\right\}\mathcal{G}^{\mathrm{TT}}(z)\mathring{\mathfrak{R}}_k^{\mathrm{TT}}(z)\right.$$
$$\left. + \left\{1 + 2g_{R^2}z^2\mathcal{G}^{\mathrm{Tr}}(z)\right\}\mathcal{G}^{\mathrm{Tr}}(z)\mathring{\mathfrak{R}}_k^{\mathrm{Tr}}(z) + \left\{2 - 11z\,\mathcal{G}^{\mathrm{c}}(z)\right\}\mathcal{G}^{\mathrm{c}}(z)\mathring{\mathfrak{R}}_k^{\mathrm{c}}(z)\right]. \quad (122)$$

The terms with more than one power of the propagator come from genuine interaction terms. In the graviton sector, they are proportional to the higher order couplings due to our regulator choice (86).

Next, we will present the flow equations for the fourth order couplings. A general feature of higher order couplings is that if the dimension is at or below the order, their flow equations feature non-integral terms. These stem from positive powers of the heat kernel expansion parameter, which map to derivatives of the function that is traced over, evaluated at zero argument. The flow of the $R^2$ coupling is

$$\dot{g}_{R^2} = \frac{\dot{g}}{g}g_{R^2} - \frac{1}{2}\frac{96\pi g}{16\pi^2}\left\{\left(4 - \frac{\dot{g}}{g}\right)\left(\frac{175}{108}\frac{\mathcal{R}_k^{\mathrm{TT}}(0)}{\mathcal{R}_k^{\mathrm{TT}}(0) - 2\lambda} + \frac{1}{72}\frac{\mathcal{R}_k^{\mathrm{Tr}}(0)}{\mathcal{R}_k^{\mathrm{Tr}}(0) - \frac{4}{3}\lambda} - \frac{1}{36}\right)\right.$$
$$+ \int_0^\infty \mathrm{d}z\left[\mathcal{G}^{\mathrm{TT}}(z)^2\mathring{\mathfrak{R}}_k^{\mathrm{TT}}(z)\left\{-\frac{20}{81}\left(1 + \mathcal{R}_k^{\mathrm{TT}\prime}(z)\right)\right.\right.$$
$$-\frac{5z}{162}\left(147g_{C^2} - 73g_{R^2} + 8\mathcal{R}_k^{\mathrm{TT}\prime\prime}(z)\right) - \frac{5z^3}{162}g_{R^2}^2\mathcal{G}^{\mathrm{Tr}}(z)$$
$$+ \left(\frac{20}{81}\left(1 + \mathcal{R}_k^{\mathrm{TT}\prime}(z)\right)^2 + \frac{80}{81}g_{C^2}z\left(1 + \mathcal{R}_k^{\mathrm{TT}\prime}(z)\right)\right) \quad (123)$$
$$\left. + \frac{5z^2}{18}\left(19g_{C^2}^2 - 10g_{C^2}g_{R^2} + 2g_{R^2}^2\right)\right)z\,\mathcal{G}^{\mathrm{TT}}(z)\right\}$$
$$+ \mathcal{G}^{\mathrm{Tr}}(z)^2\mathring{\mathfrak{R}}_k^{\mathrm{Tr}}(z)z\,g_{R^2}\left\{\frac{1}{18} + z^2g_{R^2}\left(-\frac{5}{162}\mathcal{G}^{\mathrm{TT}}(z) + \frac{1}{9}\mathcal{G}^{\mathrm{Tr}}(z)\right)\right\}$$
$$\left.\left. + \mathcal{G}^{\mathrm{c}}(z)^2\mathring{\mathfrak{R}}_k^{\mathrm{c}}(z)\left\{\frac{19}{54} - \frac{185}{162}z\,\mathcal{G}^{\mathrm{c}}(z)\right\}\right]\right\}.$$

For the flow of the $C^2$ coupling, we find

$$
\begin{aligned}
\dot{g}_{C^2} = {} & \frac{\dot{g}}{g} g_{C^2} + \frac{1}{2} \frac{32\pi g}{16\pi^2} \left\{ \left( 4 - \frac{\dot{g}}{g} \right) \left( -\frac{17}{180} \frac{\mathcal{R}_k^{\mathrm{TT}}(0)}{\mathcal{R}_k^{\mathrm{TT}}(0) - 2\lambda} + \frac{1}{120} \frac{\mathcal{R}_k^{\mathrm{Tr}}(0)}{\mathcal{R}_k^{\mathrm{Tr}}(0) - \frac{4}{3}\lambda} - \frac{7}{60} \right) \right. \\
& + \int_0^\infty \mathrm{d}z \left[ \mathcal{G}^{\mathrm{TT}}(z)^2 \mathring{\mathfrak{R}}_k^{\mathrm{TT}}(z) \left\{ -\frac{40}{27} \left( 1 + \mathcal{R}_k^{\mathrm{TT}\prime}(z) \right) \right. \right. \\
& \qquad\qquad - \frac{5z}{54} \left( 15 g_{C^2} - 2 g_{R^2} + 16 \mathcal{R}_k^{\mathrm{TT}\prime\prime}(z) \right) - \frac{5z^3}{27} g_{R^2}^2 \mathcal{G}^{\mathrm{Tr}}(z) \\
& \qquad\qquad + \left( \frac{40}{27} \left( 1 + \mathcal{R}_k^{\mathrm{TT}\prime}(z) \right)^2 \right. \\
& \qquad\qquad\qquad \left. + \frac{160}{27} g_{C^2} z \left( 1 + \mathcal{R}_k^{\mathrm{TT}\prime}(z) \right) + \frac{65}{6} g_{C^2}^2 z^2 \right) z \, \mathcal{G}^{\mathrm{TT}}(z) \Big\} \\
& \qquad\qquad - \frac{5z^3}{27} g_{R^2}^2 \mathcal{G}^{\mathrm{Tr}}(z)^2 \mathcal{G}^{\mathrm{TT}}(z) \mathring{\mathfrak{R}}_k^{\mathrm{Tr}}(z) \\
& \qquad\qquad \left. \left. + \mathcal{G}^{\mathrm{c}}(z)^2 \mathring{\mathfrak{R}}_k^{\mathrm{c}}(z) \left\{ \frac{17}{18} - \frac{91}{54} z \, \mathcal{G}^{\mathrm{c}}(z) \right\} \right] \right\} .
\end{aligned}
\tag{124}
$$

Finally, the flow of the coupling to the Euler term reads

$$
\begin{aligned}
\dot{g}_{\mathfrak{E}} = {} & \frac{\dot{g}}{g} g_{\mathfrak{E}} - \frac{1}{2} \left( \dot{g}_{C^2} - \frac{\dot{g}}{g} g_{C^2} \right) \\
& + \frac{1}{2} \frac{16\pi g}{16\pi^2} \left( 4 - \frac{\dot{g}}{g} \right) \left( \frac{103}{270} \frac{\mathcal{R}_k^{\mathrm{TT}}(0)}{\mathcal{R}_k^{\mathrm{TT}}(0) - 2\lambda} + \frac{1}{180} \frac{\mathcal{R}_k^{\mathrm{Tr}}(0)}{\mathcal{R}_k^{\mathrm{Tr}}(0) - \frac{4}{3}\lambda} + \frac{11}{180} \right) .
\end{aligned}
\tag{125}
$$

Since the Euler characteristic is a topological invariant in $d = 4$, its coupling does not appear in any of the flow equations, except in the scaling term in (125). Intriguingly, the flow of the Euler coupling can be written in terms of the other flows and a term without an integral.

The *complete* set of flow equations (123), (124) and (125) has been obtained *for the first time* without using a special background [17, 82, 162–165] (thereby neglecting some of the couplings) or expanding in some of the couplings [20, 166–168].

We note in passing that the above equations do not reduce to the standard one-loop result of perturbative Stelle gravity if the Einstein-Hilbert part of the action is neglected. There are several reasons for this. First, both the gauge fixing and the regulator are constructed with a non-perturbative setting in mind (meaning that all terms in the action are assumed to be non-vanishing), as they include an explicit factor of $1/G_N$. Thus to be able to probe the perturbative Stelle limit where $G_N \to \infty$, both would need a very careful rescaling. Second, and more importantly, we have used the background field approximation. It is well-known that this can introduce artefacts even into universal one-loop beta functions [149]. To resolve this issue, either the corresponding Ward identities have to be solved, or the limit $k \to 0$ has to be taken. Both options go beyond the scope of the present work. It is however noteworthy that with the standard higher derivative gauge fixing which makes the fourth order kinetic term minimal, the universal one-loop beta functions come out in the usual way, see *e.g.* [166–168]. This leads to the conjecture that such minimal gauge fixings might generally not need Ward identities to obtain such a result. It would be interesting to understand which classes of gauges share this property.

### 7.3 Intermezzo: fixed point search strategy

Before we move on to discuss the fixed point structure of the theory, we will present a strategy to search for fixed points in an extended truncation that are continuously connected to a fixed

point found in a smaller truncation. We illustrate the strategy going from one to two couplings, but the method is applicable generically.

Let us assume that we start with a truncation with a single coupling $g_1$, and that we have found a fixed point for it, say $g_1^*$, so that

$$\dot{g}_1(g_1^*) = 0 \,. \tag{126}$$

Now we enhance the truncation by adding a coupling $g_2$ which was set to zero before. That is, the above condition for the beta function of the coupling $g_1$ now reads

$$\dot{g}_1(g_1^*, g_2 = 0) = 0 \,, \tag{127}$$

while the beta function of the new coupling at this point in theory space is in general non-zero,

$$\dot{g}_2(g_1^*, 0) \neq 0 \,. \tag{128}$$

The search strategy is then to change the new coupling $g_2$ by a small amount, say $\epsilon$, and search for a fixed point in $g_1$ for this new value of $g_2$. Let us assume that we find a zero of $\dot{g}_1$ at the value $g_1'$, so that

$$\dot{g}_1(g_1', \epsilon) = 0 \,. \tag{129}$$

The beta function of $g_2$ will now also have changed. We then repeatedly change $g_2$ by a small amount, find a fixed point for $g_1$, and compute $\dot{g}_2$. The result is that we get $\dot{g}_2$ as a function of $g_2$ on a partial fixed point. Clearly, if we find a $g_2$ such that $\dot{g}_2 = 0$, we have found a fixed point of the complete system which is continuously connected to the fixed point which only involves $g_1$.

While this strategy will in general not find all fixed points, one can extend its applicability by starting at any value of the new coupling, search for a fixed point in the old couplings, and start the procedure from there. This gives an efficient search strategy for fixed points in all of theory space.

## 7.4 Fixed point analysis

We now study the fixed point structure of the quartic order of the derivative expansion. To set the stage, we briefly present the fixed point at the quadratic order for our setup. We then add either of the couplings individually, and finally discuss the complete system. In all of the following discussion, we choose the regulator shape function

$$\mathcal{R}_k(x) = e^{-x} \,, \tag{130}$$

that is a simple exponential regulator.

**Einstein-Hilbert truncation**   The system with $g_{C^2} = g_{R^2} = 0$ has a single fixed point at the coordinates

$$g^* = 0.534 \,, \qquad \lambda^* = 0.121 \,. \tag{131}$$

The critical exponents at this fixed point are

$$\theta_{1,2} = 2.99 \pm 1.38\mathbf{i} \,. \tag{132}$$

Since the real part is positive, the fixed point is fully attractive. This is the well-known Reuter fixed point, and the results for the critical exponents are in reasonable agreement with results published in the literature [14, 15, 23, 26, 28, 153, 169–173] when factoring in different regularisation schemes and choices of gauge fixing.

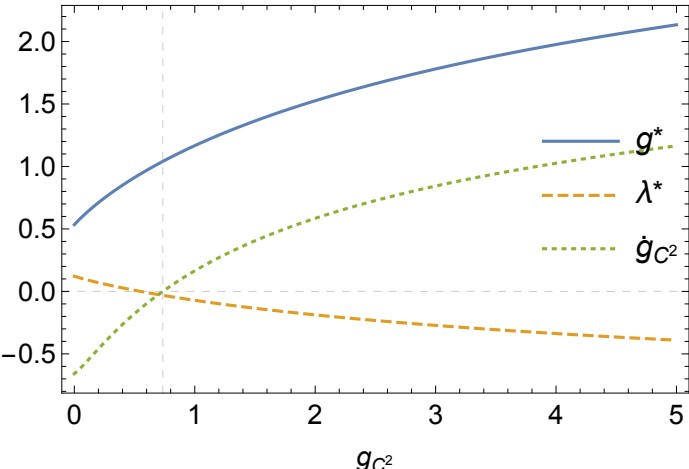

Figure 1: The partial fixed points for $g$ (solid blue line) and $\lambda$ (dashed orange line) as well as the beta function $\dot{g}_{C^2}$ (dotted green line) evaluated on the partial fixed point, as a function of the coupling $g_{C^2}$. The dashed grey horizontal line indicates zero, whereas the dashed grey vertical line indicates the combined fixed point.

**$C^2$ truncation**   We now add the $C^2$ term to the system, and use the strategy outlined in subsection 7.3 to search for fixed points. Figure 1 depicts the result of this strategy. It shows the partial fixed point values for $g$ and $\lambda$ at a range of values of $g_{C^2}$, and the value of $\dot{g}_{C^2}$ at these coordinates. The horizontal dashed line indicates zero, whereas the vertical dashed line is at the value of $g_{C^2}$ where its beta function vanishes. We find a single fixed point at

$$g^* = 1.04, \qquad \lambda^* = -0.0313, \qquad g_{C^2}^* = 0.735, \tag{133}$$

with critical exponents

$$\theta_1 = 3.83, \qquad \theta_2 = 1.85, \qquad \theta_3 = -0.953. \tag{134}$$

The value of $g_{C^2}^*$ is positive, so that no new poles arise for positive squared Euclidean momenta in the spin two sector. The inclusion of the $C^2$ term thus has two effects: first, it makes the formerly complex conjugate critical exponents real, and second, it adds an irrelevant direction, so that the value of one of the couplings can be predicted. Considering the magnitude of the critical exponents, we find a slight to moderate reduction compared to the canonical scaling dimensions, which are $4, 2$ and $0$, respectively. This is in line with the conjecture of "near-Gaussian scaling exponents" [19, 21, 43, 47, 53].

**$R^2$ truncation**   As a next step, we add the $R^2$ term to the Einstein-Hilbert system. Applying the fixed point search strategy for positive $g_{R^2}$ does not yield a fixed point, see figure 2. The slope near $g_{R^2} = 0$ indicates that there might be a fixed point for negative $g_{R^2}$ though. For this case, we however have to flip the sign of the trace regulator,

$$\mathcal{R}_k^{\mathrm{Tr}} \mapsto -\mathcal{R}_k^{\mathrm{Tr}}, \tag{135}$$

since the coupling of the highest order term in a derivative expansion dictates the sign of the regulator. As a consequence, we introduce a new singularity into the flow, which sits at

$$\lambda_{\mathrm{sing}} = -\frac{3}{4}. \tag{136}$$

We are thus confined to the region $\lambda \in (-3/4, 1/2)$, so that all propagators have the correct sign and integrals over the loop momentum exist.

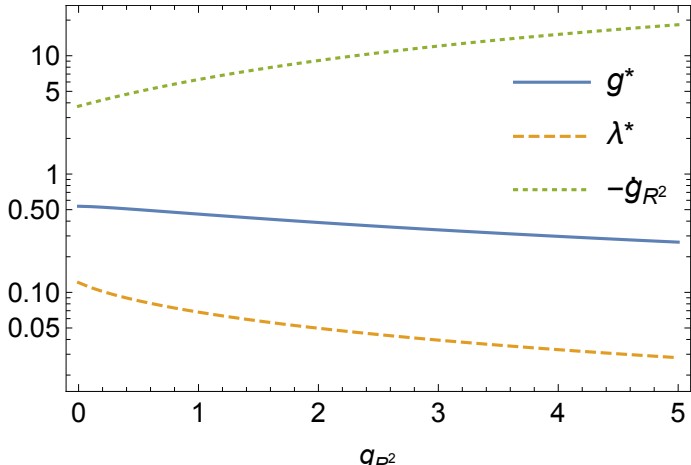

Figure 2: The partial fixed points for $g$ (solid blue line) and $\lambda$ (dashed orange line) as well as the beta function $\dot{g}_{R^2}$ (dotted green line) evaluated on the partial fixed point, as a function of positive $g_{R^2}$. The dashed grey horizontal line indicates zero. No combined fixed point is found in this regime.

This also causes the fixed point search for negative $g_{R^2}$ to not continuously connect to the case of vanishing coupling. We thus have to start our fixed point search strategy at a finite, negative value of $g_{R^2}$, search a fixed point in $g$ and $\lambda$, and apply the recursive strategy from that point. The result of this procedure is shown in figure 3, and we indeed find a fixed point at

$$g^* = 0.739, \qquad \lambda^* = 0.382, \qquad g_{R^2}^* = -1.84, \tag{137}$$

with critical exponents

$$\theta_{1,2} = 4.44 \pm 7.06\mathbf{i}, \qquad \theta_3 = -3.09. \tag{138}$$

This fixed point is much closer to the singular line $\lambda = 1/2$, and the deviation of the critical exponents from the canonical scaling is large. Moreover, the complex conjugate pair has a large imaginary part. Similar signs of instability have been observed previously in $f(R)$-type truncations [19, 21, 23, 29, 32, 34, 36, 39, 43, 47, 51–53, 110]. Including higher orders tends to tame these stronger variations.

**Complete quartic order**    Having studied the two quartic terms individually, we now set out for the full system. There are two starting points to initialise our search strategy. Starting from the fixed point with finite $g_{C^2}$, (133), we do not find a continuously connected fixed point for positive $g_{R^2}$, mimicking the case for vanishing $g_{C^2}$. On the other hand, we do find a fixed point when starting from (137). It sits at

$$g^* = 0.955, \qquad \lambda^* = 0.496, \qquad g_{C^2}^* = 0.816, \qquad g_{R^2}^* = -4.53, \tag{139}$$

and has the critical exponents

$$\theta_1 = 9.47, \qquad \theta_{2,3} = -89.6 \pm 120\mathbf{i}, \qquad \theta_3 = -162. \tag{140}$$

Judging from the very non-canonical values of the critical exponents, one might conclude that this fixed point is either an artefact of the truncation, or that higher order terms should have a large impact to stabilise the fixed point. It is likely that these extreme values arise due to the fixed point of the cosmological constant lying so extremely close to the singular line $\lambda = 1/2$. It is conceivable that higher order terms indeed shift it to smaller values, yielding more realistic critical exponents in the process, but in the end only an actual computation can give certainty about this.

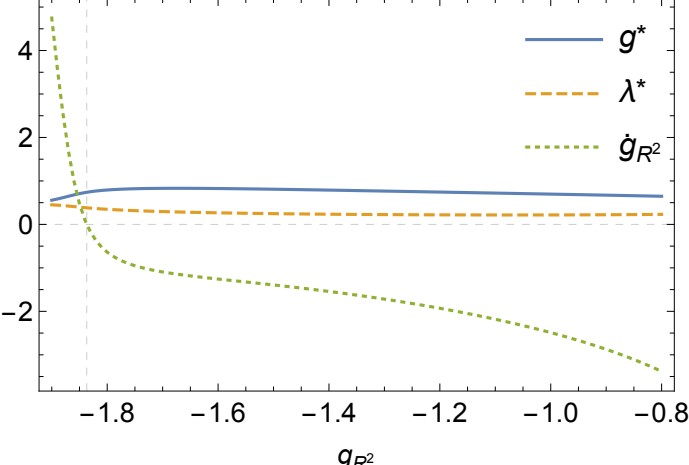

Figure 3: The partial fixed points for $g$ (solid blue line) and $\lambda$ (dashed orange line) as well as the beta function $\dot{g}_{R^2}$ (dotted green line) evaluated on the partial fixed point, as a function of negative $g_{R^2}$. The dashed grey horizontal line indicates zero, whereas the dashed grey vertical line indicates the combined fixed point.

## 7.5 Comparison: regulator without endomorphism

To investigate the stability of the results under changes of the regularisation, we briefly discuss the fixed point structure of the same system where we leave out the endomorphism in the traceless regulator. This entails using $\bar{\Delta}$ instead of $\triangle_2$ as an argument in (86). Stable fixed points should only be quantitatively affected by this.

At the Einstein-Hilbert level, and with only the $C^2$ term, this is indeed the case. In the former case, we find a fixed point at

$$g^* = 0.640, \qquad \lambda^* = 0.169, \tag{141}$$

with critical exponents

$$\theta_{1,2} = 2.78 \pm 2.14\mathbf{i}. \tag{142}$$

The critical exponents in this case are very close to the ones found above, see (132). With $g_{C^2} \neq 0$, the fixed is at

$$g^* = 1.85, \qquad \lambda^* = -0.185, \qquad g_{C^2}^* = 1.36, \tag{143}$$

with critical exponents

$$\theta_1 = 4.19, \qquad \theta_2 = 0.998, \qquad \theta_3 = -0.619. \tag{144}$$

This is in qualitative agreement with (134), and serves as an estimate of the truncation error.

As soon as we include the coupling of the $R^2$ term, we do not find a fixed point, neither without nor with finite $g_{C^2}$. This indicates again the instability mentioned earlier, and we expect more stable results when higher order terms are included.

Let us finally note that the flow of the Euler coupling within this regularisation scheme is not of the simple form (125). It rather has an additional integral, which in particular depends on $g_{C^2}$. Taking the simplicity of (125) as a guiding principle, this might be taken as an a posteriori argument for the regularisation choice (86).

## 7.6 The flow of the Gauss-Bonnet term

As mentioned earlier, the Gauss-Bonnet term is a topological invariant. Consequently, the right-hand side of the flow equation is independent of the coupling constant $g_{\mathfrak{E}}$ in front of it. This implies, absent some unlikely cancellations, that this coupling will never have a fixed point, since its beta function, evaluated on the fixed point of all other couplings, is a constant:

$$\mathfrak{e} \equiv \dot{g}_{\mathfrak{E}}\big|_{g_i = g_i^*} = \frac{2g^*}{\pi}\left(\frac{103}{270}\frac{\mathcal{R}^{\mathrm{TT}}(0)}{\mathcal{R}^{\mathrm{TT}}(0) - 2\lambda^*} + \frac{1}{180}\frac{\mathcal{R}^{\mathrm{Tr}}(0)}{\mathcal{R}^{\mathrm{Tr}}(0) - \frac{4}{3}\lambda^*} + \frac{11}{180}\right) \neq 0\,. \tag{145}$$

This can be seen very easily from the explicit form of the beta function, (125).

Depending on the sign of the constant $\mathfrak{e}$, $g_{\mathfrak{E}}$ will go to either a positive or negative infinite value at high energies. This has an interesting consequence for the weight of different topologies in the (Euclidean) path integral. Disregarding the subtleties of the reconstruction problem [174–176], if we find that $\mathfrak{e}$ is positive (negative), spacetimes with a negative (positive) Euler characteristic are enhanced, while spacetimes with a positive (negative) Euler characteristic will be suppressed. This mimics the idea of finite action [177], but the origin is the coupling instead of a divergent curvature invariant.

Due to the extremely simple structure of the beta function, we can make analytical statements about the sign of it at a fixed point of the other couplings. As a matter of fact, at our level of truncation, and with the normalisation condition that the regulators at vanishing argument are $\pm 1$ (depending on the sign of the quartic couplings), we find

$$\mathfrak{e} > 0\,. \tag{146}$$

From this it follows that generically, at this level of the truncation, the beta function for the Euler coupling is positive at the fixed point. This suggests that manifolds with a complicated topology contribute most to the Euclidean formulation of Asymptotic Safety. Incidentally, the same conclusion holds for Stelle gravity at the asymptotically free fixed point.

We illustrate this with numerical results. At the level of the Einstein-Hilbert truncation, we find

$$\mathfrak{e}^{\mathrm{EH}} = 0.523\,, \tag{147}$$

for the fixed point with finite $C^2$ coupling, we have

$$\mathfrak{e}^{C^2} = 0.282\,, \tag{148}$$

for the fixed point with negative $R^2$ coupling, the value is

$$\mathfrak{e}^{R^2} = 2.88\,, \tag{149}$$

whereas finally, in the full system, we find

$$\mathfrak{e}^{R^2 + C^2} = 28.4\,. \tag{150}$$

For a Lorentzian path integral, one might argue that either sign suppresses spacetimes with non-vanishing Euler characteristic, since they "oscillate away". This would give a dynamical mechanism whereby only spacetimes with vanishing Euler characteristic will contribute to the path integral.

# 8   Conclusion

In this paper we have set up a systematic framework to study the derivative expansion of non-perturbative renormalisation group flows in quantum gravity. We proposed a set of criteria that well-defined flows and regulators should satisfy. Then we set out to construct a suitable regulator that fulfils these criteria. Geometric considerations guided this search. As a simple example, we discussed the case of vector fields first, and found a natural way to regularise them. Moving on to gravity, we encountered some difficulties which obstruct a similar regularisation. We then used input from the action of General Relativity to nevertheless set up a well-motivated regularisation scheme. From the discussion, it is clear that our approach is applicable to any order in the derivative expansion, including an expansion in form factors.

Having set up the formal structure, we then discussed some techniques that help in practical computations and allow for an efficient evaluation of renormalisation group flows via tensor algebra software. In particular, we presented a general algorithm to obtain the propagator from an arbitrary two-point function. On the more technical side, we provided commutator rules that can be implemented generically in computer code, and discussed simplifications for tensorial terms at the maximum considered order.

We finally applied all these methods to derive and analyse the non-perturbative flow equations at the quartic order of the derivative expansion in quantum gravity. The complete set of flow equations (121) - (125) has been presented for the first time without further assumptions on top of the truncation itself. After a brief generic discussion of our fixed point search strategy, we discussed the resulting fixed point structure at different levels of sophistication. In all approximations, we find an interacting fixed point. The inclusion of the $R^2$ term introduces previously observed instabilities which are expected to be resolved by the inclusion of higher order terms. Lastly, we discussed the flow of the Euler term, which we found to flow to positive infinity at the fixed point. This indicates that Euclidean Asymptotic Safety is dominated by manifolds with negative Euler characteristic. We speculated that in Lorentzian signature, only spacetimes with vanishing Euler characteristic would contribute. This includes the flat spacetime, and gives a dynamical principle to discard more exotic structures.

Several future directions are available from here. Clearly, to resolve whether the instability of the $R^2$ term persists upon improving the truncation, the derivative expansion should be extended to sextic order, see [178] for results in the conformally reduced case. We will report results on the full case elsewhere. Another road is the inclusion of form factors along the lines of [139, 179], and discuss aspects of unitarity and causality of the theory *e.g.* in the context of scattering amplitudes [180, 181]. This would also allow to compute spectral functions from the fully momentum-dependent background propagators [140].

In any improved truncation, it will moreover be interesting whether the simple form of the flow equation (125) for the Euler coupling remains to be valid. If this would be the case, this would give a constructive proof of our observation on which topologies contribute to the path integral.

# Acknowledgements

I would like to thank Stefan Lippoldt and Alessia Platania for interesting discussions during different stages of this project, and Chris Ripken for useful comments on the manuscript.

**Funding information**   The author acknowledges support by Perimeter Institute for Theoretical Physics. Research at Perimeter Institute is supported in part by the Government of Canada through the Department of Innovation, Science and Economic Development and by

the Province of Ontario through the Ministry of Colleges and Universities.

## A   Some commutator formulas

In this appendix we prove some useful commutator formulas. The aim is to rewrite the multicommutator into a form slightly more useful for an implementation in a computer code. In the following, $X$ shall be any operator.

First, we have

$$[\Delta^l, X] = \sum_{k=0}^{l-1} \Delta^k [\Delta, X] \Delta^{l-k-1}, \qquad l \geq 1, \tag{A.1}$$

which we prove by induction. The base case $l = 1$ is trivially seen to be true. For the induction step, assume that the formula is correct for some $l \geq 1$, and calculate

$$
\begin{aligned}
[\Delta^{l+1}, X] &= \Delta[\Delta^l, X] + [\Delta, X]\Delta^l \\
&= \Delta \sum_{k=0}^{l-1} \Delta^k [\Delta, X] \Delta^{l-k-1} + [\Delta, X]\Delta^l \\
&= \sum_{k=1}^{l} \Delta^k [\Delta, X] \Delta^{l-k} + [\Delta, X]\Delta^l \\
&= \sum_{k=0}^{l} \Delta^k [\Delta, X] \Delta^{l-k}.
\end{aligned}
\tag{A.2}
$$

In the first line we used the product formula for the commutator, in the second line we used the induction hypothesis, in the third line we relabelled the sum index, and finally we combined all terms into a single sum. This establishes the claim (A.1).

The second formula that we will prove is that

$$[\Delta, X]_l = \sum_{m=0}^{l} \binom{l}{m} (-1)^m \Delta^{l-m} X \Delta^m = \sum_{m=0}^{l} \binom{l}{m} (-1)^{l-m} \Delta^m X \Delta^{l-m}, \qquad l \geq 0. \tag{A.3}$$

The equality between the two sums follows by a relabelling of the summation index. Once again we will prove this formula by induction. The base case $l = 0$ is true by the definition of

the multicommutator. Assume now that the formula holds for some $l \geq 0$ and calculate

$$
\begin{aligned}
[\Delta, X]_{l+1} &= [\Delta, [\Delta, X]_l] \\
&= \sum_{m=0}^{l} \binom{l}{m} (-1)^m \Delta^{l-m} [\Delta, X] \Delta^m \\
&= \sum_{m=0}^{l} \binom{l}{m} (-1)^m \Delta^{(l+1)-m} X \Delta^m - \sum_{m=0}^{l} \binom{l}{m} (-1)^m \Delta^{l-m} X \Delta^{m+1} \\
&= \sum_{m=0}^{l} \binom{l}{m} (-1)^m \Delta^{(l+1)-m} X \Delta^m + \sum_{m=0}^{l} \binom{l}{m} (-1)^{m+1} \Delta^{(l+1)-(m+1)} X \Delta^{m+1} \\
&= \sum_{m=0}^{l} \binom{l}{m} (-1)^m \Delta^{(l+1)-m} X \Delta^m + \sum_{m=1}^{l+1} \binom{l}{m-1} (-1)^m \Delta^{(l+1)-m} X \Delta^m \\
&= \sum_{m=1}^{l} \left\{ \binom{l}{m} + \binom{l}{m-1} \right\} (-1)^m \Delta^{(l+1)-m} X \Delta^m + \Delta^{l+1} X + (-1)^{l+1} X \Delta^{l+1} \\
&= \sum_{m=1}^{l} \binom{l+1}{m} (-1)^m \Delta^{(l+1)-m} X \Delta^m + \Delta^{l+1} X + (-1)^{l+1} X \Delta^{l+1} \\
&= \sum_{m=0}^{l+1} \binom{l+1}{m} (-1)^m \Delta^{(l+1)-m} X \Delta^m .
\end{aligned}
\tag{A.4}
$$

The first step uses the definition of the multicommutator. In the second step, we use the induction hypothesis. Afterwards, we use the definition of the commutator to split the sum into two. The next two steps implement a relabelling of the second sum. Then, we combine the overlapping parts of the two sums, then use a standard identity for the binomial coefficients. In the final step, we recombine the individual summands into the final sum. This proves the formula (A.3).

Finally, we will combine (A.1) and (A.3) and show that

$$
[\Delta, X]_l = \sum_{k=0}^{l-1} (-1)^{l+1-k} \frac{(l-1)!}{(l-k-1)! k!} \Delta^k [\Delta, X] \Delta^{l-k-1}, \qquad l \geq 1 .
\tag{A.5}
$$

We will prove this formula by direct calculation, starting from (A.3):

$$
\begin{aligned}
[\Delta, X]_l &= \sum_{m=0}^{l} \binom{l}{m} (-1)^{l-m} \Delta^m X \Delta^{l-m} \\
&= (-1)^l X \Delta^l + \sum_{m=1}^{l} \binom{l}{m} (-1)^{l-m} \Delta^m X \Delta^{l-m} \\
&= (-1)^l X \Delta^l + \sum_{m=1}^{l} \binom{l}{m} (-1)^{l-m} ([\Delta^m, X] + X \Delta^m) \Delta^{l-m} \\
&= (-1)^l X \Delta^l + \sum_{m=1}^{l} \binom{l}{m} (-1)^{l-m} X \Delta^l + \sum_{m=1}^{l} \binom{l}{m} (-1)^{l-m} [\Delta^m, X] \Delta^{l-m} \\
&= \sum_{m=1}^{l} \binom{l}{m} (-1)^{l-m} [\Delta^m, X] \Delta^{l-m} .
\end{aligned}
\tag{A.6}
$$

We started by using (A.3). We then have split off the first term of the sum, introduced a commutator and calculated one of the sums which cancelled with the first term. We also

assumed $l \geq 1$ for this in order for the splitting to make sense. We can now use (A.1) since that formula applies for the involved commutator as the sum starts at one:

$$
\begin{aligned}
[\Delta, X]_l &= \sum_{m=1}^{l} \binom{l}{m}(-1)^{l-m}[\Delta^m, X]\Delta^{l-m} \\
&= \sum_{m=1}^{l} \binom{l}{m}(-1)^{l-m}\left(\sum_{k=0}^{m-1}\Delta^k[\Delta, X]\Delta^{m-k-1}\right)\Delta^{l-m} \\
&= \sum_{m=1}^{l}\sum_{k=0}^{m-1} \binom{l}{m}(-1)^{l-m}\Delta^k[\Delta, X]\Delta^{l-k-1} \\
&= \sum_{k=0}^{l-1}\sum_{m=k+1}^{l} \binom{l}{m}(-1)^{l-m}\Delta^k[\Delta, X]\Delta^{l-k-1} \\
&= \sum_{k=0}^{l-1}(-1)^{l+1-k}\frac{(l-1)!}{(l-k-1)!k!}\Delta^k[\Delta, X]\Delta^{l-k-1}.
\end{aligned}
\tag{A.7}
$$

Here we have exchanged the order of the sums to be able to perform one of them. This completes the proof of the formula (A.5).

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
