# Peer review of "The derivative expansion in asymptotically safe quantum gravity: general setup and quartic order"

_SciPost Physics, doi:SciPost Phys. Core 4, 020 (2021)_

## Round 1 · Referee Report · Anonymous (Referee 1) · 2021-5-23

Report

This work contains technical developments for the use of the functional renormalization group in gravity, and an application to quadratic gravity.
In the first part of the paper nearly ten pages are devoted to the decomposition of vector and tensor fields in curved spacetimes. It appears however that none of this is then used in practice, since the regularization for gravity is based just on the algebraic decomposition of the metric fluctuation in its trace and traceless parts.
It is then explained how to obtain the graviton propagator, to a given order in curvature. This technique is not new, going back at least to Vilkovisky, but it may be useful to recall it here.
The original part of the paper is the application of this technique to quadratic gravity, which allows the calculation of all the beta functions without making any further approximations.
The results seem not to be stable numerically, but perhaps in line with earlier calculations.
This work must have required a nontrivial amount of calculation and it is worth reporting, but there are some points that will have to be addressed before it can be accepted for publication.

Requested changes

First, as already said, the discussion of the decomposition seems entirely unnecessary.

Second, some of the notation is inconsistent. It is quite clear from eq.(3) that z has dimensions of momentum squared. On the other hand, in section 7.2, z must be dimensionless, presumably it corresponds to z/k^2. The author should use a different symbol here. The same goes for different R's, cf. eqs.(3,23,86,129). In this connection, it would also be useful to have a more explicit formula than (86) for the regulator, and not just a "proportional to". Are the R's scalars? Its tensor structure and the prefactor 1/G, hinted after (119), should be spelled out before going to dimensionless variables.

Third, given that the results come from a lot of computer algebra, some checks are necessary. By removing many terms, it must be possible to extract the one loop results, both for GR and for quadratic gravity, and these must be compared with earlier results in the literature.

  • validity: high
  • significance: high
  • originality: high
  • clarity: high
  • formatting: excellent
  • grammar: excellent

Author:  Benjamin Knorr  on 2021-07-22  [id 1602]

(in reply to Report 2 on 2021-05-23)

[This is a literal copy of report 1, the same reply applies.]

I would like to thank the referee for their time reviewing the paper and giving constructive feedback.

1) The discussion on the decompositions is necessary, since it explains how to regularise different theories in a curved spacetime. This is a difficult problem, and no such systematic discussion has been put forward in the context of non-perturbative renormalisation group flows. The decomposition gives the mathematical justification for the specific regularisation used in the paper, the exact (partial) cancellation of gauge and ghost modes, and the decoupling of the gauge mode from the physical modes in the RG flow. The compact size of the final flow equations shows that thinking about this structure is vital - a previous study of the same system in an approximation which uses a standard recipe to define the regulator (replacing every occurrence of the Laplacian by a regularised Laplacian), ref. [168], has much more intricate flow equations.

2) I agree with the referee that the notation has not been consistent. A consistent notation has now been adopted throughout the paper. The tensor structure of the regulator shape functions is not spelled out since it follows from its argument. If the argument of the regulator shape function is tensor-valued, the expression is defined via an inverse Laplace transform or a Taylor expansion, see footnote 2.

3) The code has been tested extensively. First, it reproduces the results of ref. [168] exactly when the setup of that paper is used. This check already uses every part of the code. Second, the result, projected onto a sphere as a background, has been checked with a by-hand computation (this probes a particular linear combination of the beta functions). Third, it has been checked explicitly that the propagator computed as described in section 4 is indeed the inverse of the regularised two-point function. Fourth, the implementation of the heat kernel coefficients has been checked against the standard literature. While this is obviously not a complete proof for the correctness of the code, checks beyond this would essentially entail a complete by-hand computation of the beta functions, rendering the whole code obsolete.

When it comes to universal one-loop results, the situation is less clear. The use of the background field method is known to introduce artefacts which can spoil universal one-loop beta functions, see ref. [149], in particular section V.B. To ensure the correct result, one has to either solve corresponding Ward identities, or discuss the flow in the limit of $k\to0$. Both options are quite complicated, and go beyond the scope of the paper. On the other hand, with the standard higher derivative gauge fixing, the universal one-loop beta functions come out correctly, see e.g. ref. [168]. It would be very interesting to understand which choices of gauge fixing need corrections from the Ward identities to produce these universal results. A discussion of this point has been added to subsection 7.2.

---

## Round 1 · Referee Report · Anonymous (Referee 1) · 2021-5-23

Report

This work contains technical developments for the use of the functional renormalization group in gravity, and an application to quadratic gravity.
In the first part of the paper nearly ten pages are devoted to the decomposition of vector and tensor fields in curved spacetimes. It appears however that none of this is then used in practice, since the regularization for gravity is based just on the algebraic decomposition of the metric fluctuation in its trace and traceless parts.
It is then explained how to obtain the graviton propagator, to a given order in curvature. This technique is not new, going back at least to Vilkovisky, but it may be useful to recall it here.
The original part of the paper is the application of this technique to quadratic gravity, which allows the calculation of all the beta functions without making any further approximations.
The results seem not to be very stable numerically, but perhaps in line with earlier calculations.
This work must have required a nontrivial amount of calculation and it is worth reporting, but there are some points that will have to be addressed before it can be accepted for publication.

Requested changes

First, as already said, the discussion of the decomposition seems entirely unnecessary.

Second, some of the notation is inconsistent. It is quite clear from eq.(3) that z has dimensions of momentum squared. On the other hand, in section 7.2, z must be dimensionless, presumably it corresponds to z/k^2. The author should use a different symbol here. The same goes for different R's, cf. eqs.(3,23,86,129). In this connection, it would also be useful to have a more explicit formula than (86) for the regulator, and not just a "proportional to". Are the R's scalars? Its tensor structure and the prefactor 1/G, hinted after (119), should be spelled out before going to dimensionless variables.

Third, given that the results come from a lot of computer algebra, some checks are necessary. By removing many terms, it must be possible to extract the one loop results, both for GR and for quadratic gravity, and these must be compared with earlier results in the literature.

  • validity: high
  • significance: high
  • originality: high
  • clarity: high
  • formatting: excellent
  • grammar: excellent

Author:  Benjamin Knorr  on 2021-07-22  [id 1601]

(in reply to Report 1 on 2021-05-23)

I would like to thank the referee for their time reviewing the paper and giving constructive feedback.

1) The discussion on the decompositions is necessary, since it explains how to regularise different theories in a curved spacetime. This is a difficult problem, and no such systematic discussion has been put forward in the context of non-perturbative renormalisation group flows. The decomposition gives the mathematical justification for the specific regularisation used in the paper, the exact (partial) cancellation of gauge and ghost modes, and the decoupling of the gauge mode from the physical modes in the RG flow. The compact size of the final flow equations shows that thinking about this structure is vital - a previous study of the same system in an approximation which uses a standard recipe to define the regulator (replacing every occurrence of the Laplacian by a regularised Laplacian), ref. [168], has much more intricate flow equations.

2) I agree with the referee that the notation has not been consistent. A consistent notation has now been adopted throughout the paper. The tensor structure of the regulator shape functions is not spelled out since it follows from its argument. If the argument of the regulator shape function is tensor-valued, the expression is defined via an inverse Laplace transform or a Taylor expansion, see footnote 2.

3) The code has been tested extensively. First, it reproduces the results of ref. [168] exactly when the setup of that paper is used. This check already uses every part of the code. Second, the result, projected onto a sphere as a background, has been checked with a by-hand computation (this probes a particular linear combination of the beta functions). Third, it has been checked explicitly that the propagator computed as described in section 4 is indeed the inverse of the regularised two-point function. Fourth, the implementation of the heat kernel coefficients has been checked against the standard literature. While this is obviously not a complete proof for the correctness of the code, checks beyond this would essentially entail a complete by-hand computation of the beta functions, rendering the whole code obsolete.

When it comes to universal one-loop results, the situation is less clear. The use of the background field method is known to introduce artefacts which can spoil universal one-loop beta functions, see ref. [149], in particular section V.B. To ensure the correct result, one has to either solve corresponding Ward identities, or discuss the flow in the limit of $k\to0$. Both options are quite complicated, and go beyond the scope of the paper. On the other hand, with the standard higher derivative gauge fixing, the universal one-loop beta functions come out correctly, see e.g. ref. [168]. It would be very interesting to understand which choices of gauge fixing need corrections from the Ward identities to produce these universal results. A discussion of this point has been added to subsection 7.2.

---

## Round 1 · Referee Report · Anonymous (Referee 2) · 2021-6-24

Strengths

This paper gives beta functions for derivative expansions without further approximation, and gives interesting technical progress.

Weaknesses

  1. This paper uses code, and it is difficult to check that it is correct. Partial check is possible by comparing with earlier results.
  2. Some discussions unnecessary are included such as the decomposition of the fields.
  3. On the other hand, it does not give the details of the derivation of the main results, and only the result for specific regulator is given.

Report

This paper gives interesting technical progress in formulating functional renormalization group equation for gravity in the derivative expansion. Using mathematical code, the author gives beta functions up to quadratic order without further approximation, and I think that this is important development.

However, there are several points to be considered before accepting the manuscript. 1. The author discusses the decomposition of the fields in quite detail, but this appears not necessary in the following discussions. This part could be omitted. 2. In the presentation of the result, not much details are explained but the final result is given for a particular choice of the gauge and regulator. This is not kind to readers. The author should explain some steps to lead to his results, and also give results for arbitrary choice of the regulator, not just the result for special regulator, and then select the regulator. This way the paper gets more generality. 3. Finally the author tries to find fixed points in the set of equations, but the result is unstable numerically. This casts some doubt if the result is correct. It should be checked if the results of beta functions reproduce earlier established results, say the perturbative results of beta functions. If they agree, this gives a check, but if it does not, the author should be more cautious and discuss why this is so and/or check the results again. 4. There are several mixed notations that z is dimensionful in some equations and dimensionless in later equations. The author should give more consistent notation to avoid confusion.

Requested changes

As described in the report.

  • validity: good
  • significance: high
  • originality: high
  • clarity: ok
  • formatting: acceptable
  • grammar: good

Author:  Benjamin Knorr  on 2021-07-22  [id 1603]

(in reply to Report 3 on 2021-06-24)

I would like to thank the referee for their time reviewing the paper and giving constructive feedback.

1) This point has also been raised in reports 1/2, and the same comments apply: The discussion on the decompositions is necessary, since it explains how to regularise different theories in a curved spacetime. This is a difficult problem, and no such systematic discussion has been put forward in the context of non-perturbative renormalisation group flows. The decomposition gives the mathematical justification for the specific regularisation used in the paper, the exact (partial) cancellation of gauge and ghost modes, and the decoupling of the gauge mode from the physical modes in the RG flow. The compact size of the final flow equations shows that thinking about this structure is vital - a previous study of the same system in an approximation which uses a standard recipe to define the regulator (replacing every occurrence of the Laplacian by a regularised Laplacian), ref. [168], has much more intricate flow equations.

2) I am unsure about what the referee means. The resulting beta functions, eqs. (121) - (125), are given for arbitrary regulator shape functions (in fact, three different regulators, one each for the spin two, spin zero and ghost sector), which goes way beyond the vast majority of papers in the field. Most papers only give results for a single shape function, and in particular, often only the already evaluated threshold integrals are presented. Many of them use the so-called Litim regulator, see e.g. ref. [168] which discusses the same theory in an expansion about large Newton’s constant to linear order.

With respect to different choices of gauge, once again the paper is in line with the conventions of the field - most papers report results in a specific gauge. In particular, the paper gives a very concrete reason for the gauge choice (see the discussion in section 3) - the regularisation in this gauge is easier to set up, and there is an exact (partial) cancellation of the gauge mode and the ghost. It is also simply common that when a new level of truncation is investigated, one particularly convenient framework is chosen which makes the computation feasible in the first place. Even for the Einstein-Hilbert truncation, an exhaustive study of the gauge and parameterisation dependence only has been carried out in 2015, see ref. [28].

3) The referee presumes that the computation is incorrect based on the observation that the fixed point shows a numerical instability. Good scientific practice mandates that one should report all results, independent of whether they confirm or reject the original hypothesis (in this case that gravity is asymptotically safe). The correctness of a computation should not be challenged solely on the grounds that one did not expect the results that came out of it. In the specific case of quantum gravity, it is known that the quartic order shows some instabilities which are reduced by increasing the truncation order, see refs. [19,21,23,29,32,34,36,39,43,47,51-53,110] and as is also discussed in section 7.4. Moreover, this instability could also partially be attributed to a potential instability of the derivative expansion, as has been discussed in ref. [178] and in [2105.04566].

Regarding the reliability of the code and the results, the reply to the same question in reports 1/2 applies: The code has been tested extensively. First, it reproduces the results of ref. [168] exactly when the setup of that paper is used. This check already uses every part of the code. Second, the result, projected onto a sphere as a background, has been checked with a by-hand computation (this probes a particular linear combination of the beta functions). Third, it has been checked explicitly that the propagator computed as described in section 4 is indeed the inverse of the regularised two-point function. Fourth, the implementation of the heat kernel coefficients has been checked against the standard literature. While this is obviously not a complete proof for the correctness of the code, checks beyond this would essentially entail a complete by-hand computation of the beta functions, rendering the whole code obsolete.

When it comes to universal one-loop results, the situation is less clear. The use of the background field method is known to introduce artefacts which can spoil universal one-loop beta functions, see ref. [149], in particular section V.B. To ensure the correct result, one has to either solve corresponding Ward identities, or discuss the flow in the limit of $k\to0$. Both options are quite complicated, and go beyond the scope of the paper. On the other hand, with the standard higher derivative gauge fixing, the universal one-loop beta functions come out correctly, see e.g. ref. [168]. It would be very interesting to understand which choices of gauge fixing need corrections from the Ward identities to produce these universal results. A discussion of this point has been added to subsection 7.2.

4) This has also been raised in the reports 1/2. The notation has been made consistent.

Comments on “Weaknesses of the paper”: The referee assesses that using computer code is a weakness of the paper. Computer code is routinely used in the field to derive beta functions. For advanced studies of the RG flow, it seems unavoidable to me to use some sort of automatisation. Other fields (for example lattice gauge theory) rely completely on large scale computer simulations which can easily take months, and thus are much harder to reproduce than the results in this paper. I do not think that a by-hand computation is more accessible, as it would not be reported in the paper either, and moreover would easily span dozens of pages. The results reported in the paper are 100% based on standard, well-established techniques, and it is not obvious to me which additional details should be given to make it more accessible.

---

## Round 1 · Referee Report · Anonymous (Referee 3) · 2021-7-9

Strengths

The paper calculates effective action for quantum gravity, using derivative expansion, heat kernel methods and examines the renormalization flow equations and fixed points. The results are within the expected values, and there are some interesting observations about the Euler Characteristic flow. Equations (120)-(124) are new results, and will be useful for the examination of asymptotic scenario in quantum gravity.

Weaknesses

Some changes and discussions could improve the manuscript, in particular there is some review in the beginning, which could be abridged. A little bit more discussion on the physics of the results Equations (120)-(124) on the other hand would be appreciated. Discussions on topology change would embellish the discussion. There are some clarifications required which can make the impact of the paper better that has been given in details in below. Code used as mentioned in reference [160] should be available with the paper.

Report

The paper can be published in this journal.

Requested changes

(i) On Page 3 in itemized discussion, item 3: provide reference for previous work of instability upon inclusion of $R^2$ as mentioned.
(ii) On page 5 clarify what is meant by Laplace's operator of `unit strength'.
(iii) On Page 5 it is stated: "We rescale modes so that no nontrivial Jacobians are introduced in the Path-integral. " This has to be clarified as a non-local scaling of the fields is not a trivial scaling, this would reflect in the measure.
(iv) Equation (4) is $D_{\mu}$ a covariant derivative?
(v) Equation (27), is the operator a two point function?
(vi) Equation (38), b=1 seems to be a special point independent of the dimension d. This has to be elaborated upon, what happens if b=1 in (35), or is it not allowed to be 1?
(vii) The use of $\beta$ as a parameter as well as an index is confusing in Equation(41).
(viii) Paragraph after equation (45) the phrase `give enforce' does not make sense.
(ix) Equation (62), what is $\sigma$?
(x) Equation (111), the operator ${\cal T}$ if not defined here, has to be defined in appendix for the paper to be complete, and readable for a serious researcher.
(xi) The use of the four dimensional Euler Characteristic in Equation (113,114) is very intriguing. As mentioned earlier the use of the $R^2$ term by itself introduces instability. How does one ensure stability in a quantum situation when entered in this combination? If previous work discusses this, kindly provide the citation.
(xii) If author's code for [160] is available, that should be posted in Github as a separate file for this paper or included as an online file.
(xiii) Equation (144) if true is intriguing, and if topology `flow' is implied by the equation, it has to be elaborated. If it is not then that should be clearly stated.

  • validity: high
  • significance: high
  • originality: good
  • clarity: good
  • formatting: perfect
  • grammar: excellent

Author:  Benjamin Knorr  on 2021-07-22  [id 1604]

(in reply to Report 4 on 2021-07-09)

I would like to thank the referee for their time reviewing the paper and giving constructive feedback.

(i) The references have been added - they also appear later in the results section, but I agree with the referee that they should also be mentioned here.

(ii) Unit strength means a prefactor of one - this has been clarified in the text.

(iii) It is indeed true that rescaling fields introduces a non-trivial measure. The crucial point is however that both the decompositions and the rescalings of the decomposed fields introduce a measure, and the rescaling can be introduced in such a way to precisely cancel the measure introduced by the decomposition itself. This has been discussed in great detail in the Abelian gauge case in section 3.1, and carries over to the gravitational case.

(iv) Indeed, $D$ is a covariant derivative - this has been clarified.

(v) Equation (27) is the two-point function (that is, the second functional derivative of the action) of a free Abelian gauge field. This has been clarified.

(vi) The value $b=1$ corresponds to the case where the flat part of the operator is proportional to the transverse projector (see eq. (35)), and thus fails to possess an inverse in a derivative expansion. For the gravitational Faddeev-Popov ghost, this corresponds to the singular gauge choice $\beta=d-1$, which has been identified before, e.g. in ref. [28]. This has been clarified in footnote 3.

(vii) Some indices have been relabelled to avoid confusion.

(viii) The typo has been corrected.

(ix) As is written in the text around eq. (62), $\sigma$ is the rescaled version of $\theta$, in complete analogy to the relation of $\zeta$ and $\xi$.

(x) As mentioned in the paper, the expression of this tensor is rather lengthy. In any case, it is extremely straightforward to compute it by the methods described in the paper and basic functions of the xAct package for Mathematica. Even on an old laptop, it will not take longer than a few seconds.

(xi) I am unsure about what the referee is referring to here. The choice of the Euler characteristic in the ansatz for the effective action is simply a convenient choice of basis at the four derivative order. Any other combination of invariants which forms a basis gives the same physics output, as it corresponds to a linear redefinition of coupling constants.

(xii) The (extremely minimal) modification to the package (now ref. [161]) has been spelled out in a footnote.

(xiii) The non-vanishing of (144) is merely a consequence of the fact that the Euler characteristic is a topological invariant, so that the coupling cannot appear on the right-hand side of the flow equation. Consequently, one has 5 beta functions in 4 variables, so that the fixed point condition is over-constrained. The obvious resolution is to not require a fixed point for the topological coupling. This is unrelated to a “flow of topology”, which (if I understand the referee correctly) would imply a change of the topology with the scale $k$ - that is, with a $k$-dependent background metric. To my knowledge, this has not been looked into, but would clearly be very interesting. As far as I am aware of, the paper’s discussion about the consequences of the diverging topological coupling is new.

---

## Round 2 · Referee Report · Anonymous (Referee 2) · 2021-8-2

Report
The paper looks properly revised and I would recommend publication. However there is one point that I overlooked in my previous reading, which is about the running of the coefficient of the Gauss-Bonnet term. The author claims that the fact that the beta function for this term is independent of the coefficient and it diverges (its inverse goes to zero) is first discovered in this paper. I would like to point out that this fact is already pointed out in
K. Falls, N. Ohta and R. Percacci,
``Towards the determination of the dimension of the critical surface in asymptotically safe gravity,''
Phys. Lett. B {\bf 810} (2020), 135773 [arXiv:2004.04126 [hep-th]].
This fact should be properly cited before the publication.
K. Falls, N. Ohta and R. Percacci,
``Towards the determination of the dimension of the critical surface in asymptotically safe gravity,''
Phys. Lett. B {\bf 810} (2020), 135773 [arXiv:2004.04126 [hep-th]].
This fact should be properly cited before the publication.

---

## Round 2 · Author Response

I would like to once again thank all the referees for their time and constructive comments. The raised points have been addressed in the individual replies to the referees. Beyond the changes mentioned there, some typos have been fixed and references have been updated to include publication information.
I decided against removing some part of section 3 to keep the paper as self-contained as possible. Most of the material in this section is needed to understand the regularisation employed in this paper. The section also resolves some of the confusion in the literature, in particular regarding the trace in the transverse subspace of vector fields (see ref. [151]). Moreover it contains some explicit intermediate results like the trace of the Faddeev-Popov ghost that the referee of report 3 argues are necessary.
I decided against removing some part of section 3 to keep the paper as self-contained as possible. Most of the material in this section is needed to understand the regularisation employed in this paper. The section also resolves some of the confusion in the literature, in particular regarding the trace in the transverse subspace of vector fields (see ref. [151]). Moreover it contains some explicit intermediate results like the trace of the Faddeev-Popov ghost that the referee of report 3 argues are necessary.

---

## Round 2 · List of Changes

See the individual replies to the referees. Short summary:
- The notation has been made consistent.
- A discussion of the problems related to obtaining universal results in the background field approximation has been added.
- Some clarifications have been added where necessary, and typos have been fixed. Indices have been relabelled where they could have lead to confusion.

---

## Editorial Decision

published